

# The importance of the representation of air pollution emissions for the modeled distribution and radiative effects of black carbon in the Arctic

Jacob Schacht[1], Bernd Heinold[1], Johannes Quaas[2], John Backman[3], Ribu Cherian[2], Andre Ehrlich[2], Andreas Herber[4], Wan Ting Katty Huang[5], Yutaka Kondo[6], Andreas Massling[7], P. R. Sinha[8], Bernadett Weinzierl[9], Marco Zanatta[4], and Ina Tegen[1]

[1]Leibniz Institute for Tropospheric Research, TROPOS. Leipzig, Germany
[2]Leipzig Institute for Meteorology, Universität Leipzig, Leipzig, Germany
[3]Atmospheric Composition Research, Finnish Meteorological Institute, Helsinki, Finland
[4]Alfred Wegener Institute for Polar and Marine Research, Bremerhaven, Germany
[5]ETH Zürich, Institute for Atmospheric and Climate Science, Zurich, Switzerland
[6]National Institute for Polar Research, Tokyo, Japan
[7]Department of Environmental Science, Aarhus University, Roskilde, Denmark
[8]Department of Earth and Space Sciences, Indian Institute of Space Science Technology, Thiruvananthapura, India
[9]Aerosol Physics and Environmental Physics, Faculty of Physics, University of Vienna, Vienna, Austria

**Correspondence:** Jacob Schacht (schacht@tropos.de)

**Abstract.** Aerosol particles can contribute to the Arctic Amplification by direct and indirect radiative effects. Specifically, black carbon (BC) in the atmosphere, and when deposited on snow and sea ice, has a positive effect on the top of atmosphere radiation balance during polar day. Current climate models, however, are still struggling to reproduce Arctic aerosol conditions. We present an evaluation study with the global aerosol-climate model ECHAM6.3-HAM2.3 to examine emission-related uncer-

5 tainties in the BC distribution and the direct radiative effect of BC. The model results are comprehensively compared against latest ground and air-borne aerosol observations for the period 2005 – 2017 with focus on BC. Four different setups of air pollution emissions are tested. The simulations in general match well with the observed amount and temporal variability of near-surface BC in the Arctic. Using actual daily instead of fixed biomass burning emissions is crucial to reproduce individual pollution events but has only a small influence on the seasonal cycle of BC. Compared to commonly used fixed anthropogenic

10 emissions for the year 2000, an up-to-date inventory with transient air pollution emissions results in up to $30\%$ higher annual BC burden and an over $0.2\,\mathrm{W\,m^{-2}}$ higher annual mean all-sky net direct radiative effect of BC at top of the atmosphere over the Eastern Arctic Ocean. We estimate BC in the Arctic to lead to a net gain of up $0.8\,\mathrm{W\,m^{-2}}$ by the direct radiative effect of atmospheric BC plus the effect by an albedo reduction by BC-in-snow. Long-range transport is identified as one of the main sources of uncertainties for ECHAM6.3-HAM2.3, leading to an overestimation of BC in atmospheric layers above $500\,\mathrm{hPa}$

15 especially in summer. This is related to a misrepresentation in wet removal in one identified case at least, that was observed during the ARCTAS summer aircraft campaign. Over all, the current model version has significantly improved since previous intercomparison studies and performs now better than the AeroCom average in terms of the spatial and temporal distribution of Arctic BC.



# 1 Introduction

The near surface temperatures in the Arctic are warming at about twice the rate of the global average (Trenberth et al., 2007; Wendisch et al., 2017). Global climate models have struggled to reproduce the strength of this Arctic specific enhanced warming, that is commonly referred to as "Arctic Amplification" (AA) (Shindell, 2007; Sand et al., 2015). Aerosol particles have

the potential to substantially affect the Arctic climate by modulating the Arctic energy balance through direct and indirect radiative effects. Considering these effects in models is mandatory to reproduce the observed Arctic Amplification (Shindell et al., 2009). Within the aerosol population, black carbon (BC) is considered to be the strongest warming short lived radiative forcing agent (Quinn et al., 2015), mainly by absorption of solar radiation in the atmosphere and by reducing the albedo of snow and sea ice surfaces when deposited. The direct radiative effect of BC on the Arctic has been shown to depend on many

factors. Kodros et al. (2018) show that different assumptions about the mixing state of BC modulate the magnitude of the direct radiative effect, while its sign largely depends on the albedo of the underlying surface. Sand et al. (2012) come to the conclusion, that an increase of BC burdens in the mid latitudes could have a stronger effect on Arctic sea ice concentrations and temperatures than an increase of BC concentrations in the Arctic by modulating the meridional energy transport.

The main sources of Arctic BC are located outside of the Arctic circle, and originate mainly from fossil fuel use and biomass

burning. Local emissions exist in the forms of shipping, domestic fuel burning in remote locations, gas flaring and biomass burning (Stohl et al., 2013). With declining sea ice concentrations the emissions from local shipping are expected to increase (Corbett et al., 2010; Gilgen et al., 2018). Though human activities in Northern Russia represent an important source of BC in the Arctic, these emissions are often underrepresented in recent emission inventories, often missing gas flaring (Stohl et al., 2013; Huang et al., 2015). Gas flaring is important for the Arctic because of close vicinity (Stohl et al., 2013).

The concentration of BC and other aerosol types like organic carbon, sulfate, and dust in the Arctic is the highest in late winter/early spring and shows a minimum during the summer. The maximum is often referred to as "Arctic Haze" and is caused by the southward expansion of the Arctic front, which promotes the transport of pollutants from the mid-latitude emission zones (Law and Stohl, 2007). The Arctic front is a barrier of air with a colder potential temperature, that impedes mixing of air masses, reducing wet removal (Shaw, 1995). In summer, the northward retreat of the Arctic front combined with an intensification of

precipitation events leads to a minimum in the aerosol concentration (Law and Stohl, 2007). Koch et al. (2009) show that the observed seasonal variability in BC concentrations is challenging for global aerosol models. They showed a tendency to underestimate peak near surface BC concentrations in late winter/early spring (Shindell et al., 2008). Although more recent studies show an improvement in the representation of the high late winter/early spring concentrations (e.g. Eckhardt et al., 2015; Sand et al., 2017), the model-to-model variability of simulated BC concentration remains considerable (Eckhardt et al.,

30  2015).

Despite a good agreement between BC estimations from models and observations close to source regions (Bond et al., 2013), in remote regions models tend to predict a higher BC concentration at the surface and especially in the upper troposphere (Schwarz et al., 2013). This is caused by a misrepresentation of the aerosol removal processes and transport (Schwarz et al., 2013). The mixing and aging, as well as the related removal of aerosol particles along the various transport pathways, are



important processes that need to be described accurately in the models (Vignati et al., 2010). The representation of emissions is, however, a prerequisite to correctly simulate the transport fluxes and are therefore a key source of uncertainties (Stohl et al., 2013; Arnold et al., 2016; Winiger et al., 2017). Both the coverage of all BC sources and their temporal variability are contributing to the ability to reproduce vertical BC distributions (Stohl et al., 2013). The relative source contributions are,
however, still discussed with different results (Winiger et al., 2017)).

Having pointed out the potential importance of BC for the AA and the additional uncertainties in aerosol-climate models, in this study, we thoroughly evaluate the global aerosol-climate model ECHAM-HAM for the period 2005 to 2017 with focus on BC in the Arctic. The evaluation uses a comprehensive set of ground and airborne in-situ measurements of BC all across the Arctic and throughout all seasons. In order to address emissions as one of the main sources of uncertainty, we make use
of different emission setups to assess the sensitivity of our model to the emission data used. The emissions are composed of different state-of-the-art and widely used emission inventories of anthropogenic air pollution and wild fires. The sensitivity studies allow for estimating the uncertainty range of the BC burden and climate radiative effects in recent aerosol-climate model simulations that are related to emission uncertainties. Estimates of BC radiative effects presented in this study comprise the atmospheric radiative perturbation and the BC-in-snow albedo effect. The model results utilizing the different emission
inventories are compared among each other in such a way that the following three points can be explored: (1) The importance of considering daily varying biomass burning emissions, (2) uncertainties in current anthropogenic emission inventories, and (3) the potential improvements by regional refinements in particular in Russian air pollution sources, including gas flaring.

The methods used in this study are discussed in Section 2, with an overview over the model setup and in-situ measurements. The sensitivity and related uncertainty in the atmospheric BC burden will be explored in Section 3. Section 4 will then discuss
how well the model performs with the different setups in comparison to BC concentrations obtained by the in situ measurements. Finally, we provide an up-to-date evaluation of the direct radiative effect of BC in the Arctic region, and quantify an uncertainty range for this effect that is related to the different emissions (Section 5).

## 2 Methodology

### 2.1 Model description

For this study the global aerosol climate model ECHAM-HAM is used. It was first described in Stier et al. (2005). We used the latest version ECHAM6.3-HAM2.3 developed by the HAMMOZ community, ECHAM6-HAM2.3 (Tegen et al., 2018). The model is based on the general circulation model ECHAM, developed by the Max Planck Institute for Meteorology MPI-M in Hamburg (Stevens et al., 2013). ECHAM is online coupled to the aerosol module HAM that is described in detail in Zhang et al. (2012). It uses the aerosol mircophysics module M7 (Vignati et al., 2004; Zhang et al., 2012), in which BC, sulfate (SU),
organic carbon (OC), sea salt (SS), and mineral dust (DU) are the aerosol species that are accounted for. Volcanic emissions are prescribed. The emission fluxes of mineral dust from deserts as well as, sea salt and dimethyl sulfide (DMS) originating from the ocean are calculated online, depending on the meteorology (see Zhang et al., 2012; Tegen et al., 2018). Anthropogenic and biomass burning aerosol emissions are prescribed from emission inventories for which different setups are available.





The aerosol number concentration, as well as the mass concentration are prognostic variables calculated using a "pseudo-modal" approach. The log-normal modes represent: the nucleation mode with a dry radius ($r_{dry}$) range of 0-5 nm and a geometric standard deviation ($\sigma_{\ln(r)}$) of 1.59, Aitken mode ($r_{dry}$=5-50 nm, $\sigma_{\ln(r)}$=1.59), accumulation mode ($r_{dry}$=50-500 nm, $\sigma_{\ln(r)}$=1.59) and coarse mode ($r_{dry} >$500 nm, $\sigma_{\ln(r)}$=2.0). The latter three exist as hydrophilic and hydrophobic (commonly

referred to as soluble and insoluble, respectively). Aerosol in the nucleation mode is always considered hydrophilic, consisting solely of sulfate. The Aitken mode contains BC and OC, which, in the hydrophilic class, occur internally mixed with sulfate. The accumulation and coarse mode contains BC, OC, and DU for both classes, and SU (internally mixed), as well as SS, for the mixed classes. Aerosol particles within a mode are assumed to be internally mixed, such that each particle can consist of multiple components. Aerosols of different modes are externally mixed, meaning that they co-exist in the atmosphere as

independent particles. During the mixing, aging, and coagulation processes, that are parameterized in M7, aerosol can grow to a bigger mode and can be coated with sulfate to transfer from the hydrophobic to hydrophilic mode. The median radius of the modes can be calculated from the number and mass concentration.

The removal process in ECHAM6.3-HAM2.3 is split between sedimentation, dry deposition and wet deposition. The sedimentation process describes the removal by gravitational settling and is applied only to accumulation and coarse mode particles.

In the model, dry deposition is due to turbulent mixing and affects all but the nucleation mode particles. In the wet deposition scheme, particles are removed as activated aerosol only if the cloud is precipitating. Additionally below cloud scavenging is applied. For more details on the removal processes in ECHAM-HAM see Zhang et al. (2012). Monthly and yearly mean values of BC emissions and deposition fluxes computed by ECHAM-HAM for the Arctic are given in Table 1. Wet deposition accounts for over 90 % of the BC removal and is therefore a crucial impact factor on the Arctic BC burden.

The modeled spatial aerosol distribution affects the climate simulations through interactions with radiation and clouds. A look-up table with pre-calculated Mie parameters is used to dynamically determine the particle optical properties considering their size, composition, and water content (Stier et al., 2005; Zhang et al., 2012). The description of cloud micro-physics in ECHAM6.3-HAM2.3 is based on the two-moment scheme of Lohmann et al. (2008), which allows to account for the impact of modeled aerosol populations on the number concentrations of cloud condensation nuclei and ice nucleating particles. Particles

can collide with droplets/ice particles after they have formed. For further details on the model system we refer to Stier et al. (2005) and Zhang et al. (2012).

## 2.2 Emission inventories

While here we focus on BC, the details on the emissions of other aerosol species can be found in Zhang et al. (2012). BC is emitted only in the hydrophobic Aitken mode with a median radius of $r_{dry} = 30$ nm and can grow into the bigger modes by

aging and coagulation. It can also become hydrophilic by getting coated with sulfate. In this study we use and compare four different emission setups, that are build from different emission datasets as described in the following.

We use the emissions developed for the Atmospheric Chemistry and Climate Model Intercomparison Project (ACCMIP) as described by van Vuuren et al. (2011). The data has a $0.5° \times 0.5°$ horizontal resolution and contains anthropogenic and biomass burning emissions that do not differ between years. The ACCMIP emission inventory is available with historic emissions until




the year 2000, and for four different development scenarios, linked to the Representative Concentrations Pathways (RCPs) for all later years (2000-2100) (Lamarque et al., 2010). However, in this study we only use year 2000 emissions.

The global emission data set created for Evaluating the Climate and Air Quality Impacts of Short-lived Pollutants (ECLIPSE), version 5a by Klimont et al. (2017) includes only anthropogenic emissions. The horizontal resolution is $0.5° \times 0.5°$. Historic emissions are available until 2010 and projections of different industrial development scenarios afterwards, that are linked to the RCPs. Unlike the ACCMIP emission data set, it also includes emissions from gas flaring. However, gas flaring emissions from Northern Russia, which are supposed to play an important role for the Arctic BC level, have been discussed as being too low in current emission inventories (Stohl et al., 2013).

To better represent Russian BC emissions, we use the anthropogenic BC emission data set described in Huang et al. (2015). It has a $0.1° \times 0.1°$ horizontal resolution and is available for the year 2010 only. Since this data set is limited to the area of Russia, we combine it with the ECLIPSE emission data. The emissions of Russian gas flaring are more than 40% higher than in ECLIPSE (Huang et al., 2015). Since this data set is based on the most recent information it is taken as the best source currently available. Compared to each other, ACCMIP and ECLIPSE span a range of uncertainty regarding current estimates of anthropogenic emissions.

GFAS (Global Fire Assimilation System) is a dataset of biomass burning emissions. The strength of the emissions is scaled to the fire radiative power as observed by the MODIS instruments on board NASA's Aqua and Terra satellites (Kaiser et al., 2012). This allows for a representation of real-time fires in ECHAM-HAM and enables it to reproduce the biomass burning plumes, that are regularly observed in the Arctic during spring and summer months. This covers natural fire events, as well as those caused by anthropogenic activities.

## 2.3 Experimental setup

We run ECHAM6.3-HAM2.3 at T63 horizontal resolution (approximately $1.8°$), with 47 vertical layers. The model is driven with ERA-Interim reanalysis data and prescribed sea surface temperature (SST) as well as sea ice concentrations (SIC). The model simulations cover the 11-year period 2005-2015. One run is extended to June 2017 in order to include the period of a recent aircraft campaign. In total four model runs are realized, each with a different combination of emission datasets as described in the following. The time-averaged land emissions of BC from each setup are presented for different geographical regions in Table 2 (see Figure 1 for location).

For the first run we use the historical year 2000 ACCMIP emissions, throughout the whole simulation period. Hereafter, this run is referred to as "ACCMIP". Although fixed for the year 2000, ACCMIP emission data is still widely used for model experiments. This simplification is a common approach to reduce degrees of freedom and control boundary conditions in non-transient climate studies. ACCMIP is the only run that does not use the daily updated GFAS emissions for biomass burning and can therefore not be expected to reproduce actual biomass burning events. Therefore, it can serve as a reference run needed to estimate the uncertainty that is related to the representation of biomass burning emission. The resulting monthly BC for the latitude bands of 30-60° N and 60-90° N can be seen in Figure 2. Sulfate is important for the aging and wet removal of BC, therefor the $SO_2$ plus sulfate ($SO_X$) emissions are given as well. It is the run with the highest European emissions, with





538 kt yr$^{-1}$, and low Central Asian emissions, see Table 2. The anthropogenic ACCMIP emissions are higher in Europe than for the other data sets used in this study, because recent changes in air quality regulations led to lower emissions there in the period examined (considered until 2011). In Southeast Asia, they are however smaller, since the Asian economy has strongly grown since 2000 and with it the air pollutant emissions.

The second run, called "ACCMIP-GFAS", combines the biomass burning emissions of GFAS with the year 2000 ACCMIP emissions from anthropogenic sources (orange line in Figure 2). This run also does not account for changes in anthropogenic emissions but considers the day-to-day variability of wild fires. Together with a setup described in the following, it can be used to assess the range of uncertainty in anthropogenic emissions. This run has the highest average BC emissions in North America, with 515 kt yr$^{-1}$ and the lowest Central Asian emissions, with 1997 kt yr$^{-1}$.

In the third, run we use the ECLIPSE RCP4.5 emission data combined with GFAS emission. It is referred to as "ECLIPSE" hereafter (blue line in Figure 2). This run has the highest BC emissions in Central Asia, with about 1.5 times the emissions of the ACCMIP runs, cf. Table 2.

The fourth run, which is referred to as "BCRUS", uses the updated spatially highly resolved BC emissions from Huang et al. (2015), replacing and updating only the anthropogenic BC emissions in Russia. Elsewhere the emissions are the same as in the

ECLIPSE run. This way the BC sources are supposed to be better represented while otherwise considerably underestimated in the global datasets, in particular, with respect to gas flaring emissions. For other species, most notably SO$_X$ the runs BCRUS and ECLIPSE do not differ, see green lines in Figure 2.

BCRUS is chosen as the reference run, since it uses the most up-to-date data and is therefore assumed to be the best estimate. In BCRUS the BC emissions north of 60° N on land are even higher than over the oceans compared to the other datasets, with

172 kt yr$^{-1}$ and 7 kt yr$^{-1}$, respectively.

Figure 3a shows the emissions of BC for the reference run BCRUS. The highest emissions north of 30° N are found in the industrial regions of East Asia, Europe and North East America as well as in gas and oil extraction areas in North America and northern Russia. The anthropogenic emissions in the sparsely populated Northern Canadian and Alaskan regions are much lower than those of the densely populated European region. Additionally, the aforementioned gas flaring emissions in Russia

are assumed to be higher than in northern North America. The transport efficiency from the East Asian sources to the Arctic is comparably low but the high emission flux in this region makes it important for the Arctic by contributing to higher atmospheric layers (Ikeda et al., 2017).

Figures 3b-c show the difference in BC emissions for the runs ACCMIP-GFAS and ECLIPSE compared to the BCRUS setup, respectively. In ACCMIP-GFAS BC emissions are higher in North America, Europe, western Russia and Japan. In

northern Russia and China, however they are considerably smaller with locally of over 3500 kg km$^{-2}$ yr$^{-1}$ less than in BCRUS and over 2800 kg km$^{-2}$ yr$^{-1}$ less, respectively. Figure 3c shows the difference between ECLIPSE and BCRUS. There are only differences in Russia, as expected. The ECLIPSE emissions are smaller than the BCRUS emissions, because of newer information about additional sources. Among other sources, higher values are mainly due to gas flaring emissions. Figure 3d shows difference in BC emission between the runs ACCMIP and ACCMIP-GFAS that result from their difference in the



biomass burning representation discussed above. ACCMIP shows higher emissions in Europe and Russia, while the emissions of ACCMIP-GFAS are higher in North America. The totals of BC emissions are summarized in Table 2.

## 2.4 Calculation of direct aerosol radiative effects of BC

For diagnostic output, the instantaneous radiative impact of all aerosol types is calculated in ECHAM-HAM by calling the radiation routine twice, once considering the interaction between aerosol particles and radiation and once without any aerosol. The difference between these two calls is then considered to be the direct aerosol radiative effect (DRE), which is free of any rapid adjustment (semi-direct effects).

To calculate the DRE by BC, the ACCMIP-GFAS and BCRUS runs were repeated without BC being considered in the calculation of radiative fluxes for the sub-period 2005-2009. The DRE of BC is then derived from the difference of these two runs to their original setup. The shorter time period was chosen to save computational resources.

The aerosol transport and radiation simulations in this study consider the reduction of snow albedo due to deposited BC. The BC-in-snow albedo effect is parameterized in terms of a lookup table based on a single-layer version of the snow, ice, and aerosol radiation (SNICAR) model from Flanner et al. (2007). The scheme was first implemented in the earth-system model version of ECHAM6 by Engels (2016), and since recently is available in ECHAM6.3-HAM2.3 (Gilgen et al., 2018). It accounts for the BC concentration within in the uppermost $2\,\mathrm{cm}$ of snow. Input parameters are the snow precipitation, the sedimentation, dry deposition and wet removal of BC, as well as on snow melt and glacier runoff, the latter of which leads to an enrichment of BC in the remaining snow layer. The BC-in-snow albedo effect is computed for solar radiation only because the albedo is only used for shortwave (solar near-infrared and visible) wavelengths in the model. The effect in the terrestrial spectrum is very small and can be neglected for the atmosphere. A feature not considered so far is the impact of BC deposition on bare sea ice. This, however, is expected to be negligible since the spatial extent of sea ice without snow cover is small (Gilgen et al., 2018). In this study, the parameterization is only used for diagnostics of the BC-in-snow albedo effect without any feedback on the model dynamics.

## 2.5 Observations

### 2.5.1 Near surface BC concentrations

Near surface BC concentrations are taken from different measurement sites around the Arctic, shown on the map in Figure 4 as triangles. These sites utilize different measurement principles providing BC concentrations that differ by definition. The measurement principles, measurement period, as well as the location of the measurements are summarized in Table 3.

At five of these sites light absorption was measured with Aethalometers. Of those five stations Alert and Summit measured at $467\,\mathrm{nm}$, $525\,\mathrm{nm}$ and $637\,\mathrm{nm}$, Zeppelin Station measured at $525\,\mathrm{nm}$ only, and Tiksi and Pallas measured at $637\,\mathrm{nm}$ only. From that equivalent black carbon (eBC) concentrations were calculated using a mass absorption coefficient (MAC) $9.8\,\mathrm{m^2\,g^{-1}}$ for aged Arctic BC at $550\,\mathrm{nm}$ of (Zanatta et al., 2018), for the stations where measurements at $525\,\mathrm{nm}$ where available. For the stations where the light absorption was only available at $637\,\mathrm{nm}$, a MAC of $8.5\,\mathrm{m^2\,g^{-1}}$ for Scandinavian BC was used at





this wavelength, correspondingly (Zanatta et al., 2016). The data was processed as described in Backman et al. (2017a) to reduce noise and lower the detection limit, which is important for the Arctic since concentrations tend to be about one order of magnitude lower than at mid latitudes outside of Arctic Haze season. We use the variable collection time data from Backman et al. (2017b), that covers January 2012 until December 2014. The sites are: Alert, Nunavut, Canada; Pallas, Finland; Tiksi,

Sacha, Russia; Zeppelin, Svalbard, Norway; Summit, Greenland, Denmark.

BC concentration data measured with a continuous soot monitoring system (COSMOS), which removes volatile aerosol compounds, is available for Ny-Ålesund, Svalbard, Norway, and Barrow, Alaska, USA, for the period 1 April 2012 to 31 December 2015 and 12 August 2012 to 31 December 2015, respectively. The data collection and the retrieval of BC mass concentrations using a MAC of $8.73 \, \mathrm{m^2 \, g^{-1}}$ at $565 \, \mathrm{nm}$ is described in Sinha et al. (2017).

In addition, we use measurements of eBC concentration at Villum research station in North Greenland, that were performed with a multi-angle absorption photometer (MAAP). We use daily averaged data from 14 May 2011 to 23 August 2013. Further information on data sampling and processing can be found in Massling et al. (2015).

For Alaska we use filter collected BC data acquired by the Interagency Monitoring of Protected Visual Environments (IM-PROVE) aerosol network. The thermal protocol used to process the measurements is described in Chow et al. (2007). We use

data from the sites Tuxedni, Trapper Creek, Denali National Park (NP) and Gates of the Arctic NP.

### 2.5.2 Aircraft campaigns

The correct representation of the modeled aerosol layering is a key prerequisite for estimating the aerosol radiative impact. For this reason, we collected BC measurements from five Arctic airborne campaigns. During all flights, the mass concentration of refractory black carbon (rBC) was quantified by means of the single particle soot photometer (SP2), which ensures the high

time resolution and high sensitivity required in airborne observations.

The HIPPO (HIAPER Pole-to-Pole Observation) campaign consists of 5 deployments by the National Science Foundation (NSF) (data set: Wofsy et al. (2017), version 1): The HIPPO-1 (9 to 23 January 2009), HIPPO-2 (31 October to 22 November 2009). HIPPO-3 (24 March to 16 April 2010), HIPPO-4 (14 June to 11 July 2011) and HIPPO-5 (9 August to 8 September 2011). Flights included northern hemisphere high latitudes over North America, the Pacific Ocean and the Bering Sea. BC

particles were measured with a SP2. The aircraft used, was the NSF/NCAR Gulfstream-V (GV).

The BC data from NASA's campaign ARCTAS (Arctic Research of the Composition of the Troposphere from Aircraft and Satellites) was collected in two deployments: Spring (April 2008) and Summer (June/July 2008), over North America and the American Arctic. The mission design and execution are described in Jacob et al. (2010) (data set: SP2_DC8, https://www-air.larc.nasa.gov/cgi-bin/ArcView/arctas/).

The summer campaign of ACCESS (Arctic Climate Change, Economy and Society) in July 2012 took place over Scandinavia and the European Arctic (Roiger et al., 2015). The BC mass concentration was derived from measurements of a single particle soot absorption photometer (SP2) on board of the Falcon aircraft of the DLR (Deutsches Zentrum für Luft und Raumfahrt).

Another set of airborne measurements was collected from the 2017 PAMARCMiP (Polar Airborne Measurements and ARctic Regional Climate Model Simulation Project) campaign (Herber et al., 2012). The selected flight took place in March,



and was based in Longyearbyen, Spitzbergen, Norway and made use of the Polar 5 aircraft of the Alfred-Wegener-Institute (AWI).

Also based in Ny-Ålesund was the ACLOUD (Arctic CLoud and Observations Using airborne measurements during polar Day) campaign, with measurements from 22 May to 28 June 2017 (Wendisch et al., 2018). Again, the BC concentrations were measured with an SP2 aboard the AWI Polar 5 aircraft.

The range of flight tracks of the aircraft campaigns used in this study are mapped in Figure 4 as colored boxes, with HIPPO in blue, ACCESS in red, ARCTAS in orange and the combination of ACLOUD and PAMARCMiP-2017 in green. The most western, eastern, southern and northern coordinates, at which the aircraft took measurements, form the edges of the boxes, with measurements south of $60^{\mathrm{c}}\mathrm{irc}\,\mathrm{N}$ being not considered. An overview over instruments and dates are given in Table 3. Even though aircraft campaigns can only give information within a short time window, the combination of different campaigns allows for decent coverage except for December, February, September and October, the months for which no aircraft data is available.

## 3 Sensitivity study on emissions

In order to investigate the uncertainty range in BC burden and its direct radiative impact, that results from the uncertainty in emissions, different simulations with the aerosol climate model ECHAM6.3-HAM2.3 using four emission configuration are performed and compared as outlined in Section 2.3.

The atmospheric burden of BC averaged of the simulation period (2005-2015), that results from the different emission setups is shown in Figure 5. The distribution over the BC burden resulting from BCRUS (see Figure 5a) is comparable to the distribution of the emissions in this run (see Figure 3a). The northward transport results in a visible separation between the eastern and western hemispheres in the BC burden northern part of $60°\,\mathrm{N}$, with higher values of 200 to $800\,\mathrm{\mu g\,m^{-2}}$ on the eastern hemisphere, compared to values of 50 to $400\,\mathrm{\mu g\,m^{-2}}$ on the western hemisphere. This separation along the prime meridian is a result of higher anthropogenic emissions in the north of the eastern hemisphere, as discussed in Section 2.3. The area-weighted mean burden of BC north of $60°\,\mathrm{N}$ of BCRUS is $254\,\mathrm{\mu g\,m^{-2}}$ on the multi-year annual average, which is the highest among the model runs used for this study. The highest values north of $60°\,\mathrm{N}$ are located in the Russian gas flaring region with over $560\,\mathrm{\mu g\,m^{-2}}$.

The causes and details, as well as differences between the runs will be discussed in the following.

### 3.1 Recent economic changes

To estimate the range of anthropogenic emissions in currently widely used inventories, we compare the runs BCRUS and ACCMIP-GFAS. The ACCMIP run does not take recent economic changes into account, since emissions are fixed to the year 2000. BCRUS on the other hand is largely based on the ECLIPSE emissions that considers the economic development until 2015 and provides projections for the years after. Since both are combined with the biomass burning emissions from GFAS



(that covers natural as well as human-ignited fires), the differences in BC emissions are solely in the anthropogenic emissions (excluding human-caused grass and forest fires).

The use of fixed emissions in ACCMIP-GFAS causes a remarkable difference in the atmospheric burden of BC over the source regions compared to the reference run (see Figure 5b). ACCMIP-GFAS does not catch the reduction in BC emission

over western countries and Japan due to the implementation of strict air quality legislation and the increased emission over China caused by its economic growth. The neglecting of the recent economic evolution and mitigation policies result in an overall underestimation of the BC burden by $63\,\mu g\,m^{-2}$ (25 %) within the 60-90° N latitudinal band. Over the Kara Sea, the result is an underestimation that exceeds $100\,\mu g\,m^{-2}$, a region that has been discussed as a hot spot for the connection between Arctic sea ice loss and changes in the large scale atmospheric circulation with particular sensitivity (e.g. Petoukhov

and Semenov, 2010).

### 3.2 Regional refinement

Higher, more realistic estimates of emissions for Arctic sources (e.g., gas flaring) have been discussed as a requirement to reproduce observations like locally high BC concentrations in snow (Eckhardt et al., 2017), as well as the layering and seasonality of Arctic aerosol concentration far from sources (Stohl et al., 2013). However, improving the regional accuracy of

BC emissions in the Russian Arctic does not impact the modeled BC spatial distribution meaningfully outside of the Russian Arctic. As seen in the comparison of the runs BCRUS and ECLIPSE (Figure 5c), the difference in BC burden between BCRUS and ECLIPSE shows only differences visible in Russia, since only there the BC emissions differ (see Table 2 and Figure 3). This results in an increase in the BC burden mainly in the eastern Arctic with up to $25\,\mu g\,m^{-2}$ higher values over the Barents and Kara-Sea. The area-weighted annual averages north of 60° N differs by $11\,\mu g\,m^{-2}$, with the higher BC burden produced

by BCRUS. However, stronger effects are found for BC near surface concentrations as discussed below, due to the vicinity of the refined sources to the Arctic and the resulting transport at lowest atmospheric levels.

### 3.3 Temporal variability of wildfire events

The atmospheric composition and, in particular, the BC loading is strongly influenced by wild fires, which have a strong spatio-temporal variability. The importance of considering actual biomass burning events is demonstrated by comparing the runs

ACCMIP-GFAS and ACCMIP. While ACCMIP-GFAS accounts for real fire events derived from satellite retrievals, ACCMIP uses fixed fire emissions for the year 2000. The ACCMIP-GFAS BC emissions are higher than the ones of ACCMIP by $64.5\,kt\,yr^{-1}$, mainly caused by North American emissions (see Figure 3d).

The patterns of BC burden of both runs are similar, with a higher burden over the western industrialized countries and a lower burden over China, compared to BCRUS. The area weighted average burden of BC estimated with ACCMIP is $186\,\mu g\,m^{-2}$,

which is $11\,\mu g\,m^{-2}$ (6 %) less than ACCMIP-GFAS. A map of the differences in annual average burden of BC due to the different representations of biomass burning emissions is shown in Figure 5d. A clear pattern of lower BC burden over Southern Siberia and a higher burden over North America is visible. For the high Arctic both runs produce a similar burden in the 11-year



mean with differences in BC burden of less than $25\,\mu g\,m^{-2}$. However, for short time periods, influenced by biomass burning events the difference between the two runs can be dramatic, as shown below for comparisons of the BC mass concentration.

## 4 Evaluation with observations

### 4.1 Near-surface BC mass concentration

Near-surface measurements of BC mass concentrations can help evaluate the capability of ECHAM-HAM to reproduce the distribution of BC in the Arctic atmosphere for reasonable estimates of the warming influence of absorbing aerosol. While the data is only representative of the lowest atmospheric layer, the long time series give robust information about this specific important climate forcer. The multi-year seasonality of near-surface BC is compared to observations in the Arctic as is the temporal correlation with a spatial emphasis.

Figures 6 through 8 each show the comparison of the observed and modeled monthly median mass concentration of near-surface BC, for four available Arctic field sites averaged over multiple years. A list with detailed information on measurement period, instrumentation and data providers can be found in Table 3. The model is compared to the measurements in terms of how well the annual cycle is reproduced by comparing median BC mass concentration values, and in terms of the ability to reproduce pollution events at the correct time by analyzing correlation coefficients,

Of the stations used, Zeppelin Station and Ny-Ålesund are located on Svalbard. Alert and Villum Research Station are both situated in the north of the Greenland ice sheet. The annual cycle of the BC concentration is shown in Figure 6 in terms of the median, upper and lower quartiles in black; the different model runs are shown color coded. At all four stations, the maximum median BC mass concentration is observed in Spring, with $36\,ng\,m^{-3}$, $72\,ng\,m^{-3}$ and $73\,ng\,m^{-3}$ for Zeppelin Station, Villum Research Station and Alert in March, respectively.

For Ny-Ålesund the highest concentrations are observed in April with a median of $30\,ng\,m^{-3}$. For all stations in Figure 6 there is a minimum in summer with less than $15\,ng\,m^{-3}$ median BC concentrations in the near-surface air. At all four stations, the reference run BCRUS produces higher median concentrations in January than observed. The modeled BC mass concentrations are underestimated by the model at all of these stations except Ny-Ålesund, at least for some months. The model overestimates the BC concentrations in the beginning of the year at all stations. The overestimation is largest at Ny-Ålesund with monthly median values of up to $120\,ng\,m^{-3}$ for BCRUS, compared to the measured median of $20\,ng\,m^{-3}$.

For Zeppelin Station and Ny-Ålesund, BC is also overestimated in November and December. Here, the model simulates monthly median values each of $90\,ng\,m^{-3}$ for December compared to measured medians of $10\,ng\,m^{-3}$ and $20\,ng\,m^{-3}$ for Zeppelin Station and Ny-Ålesund, respectively. It has to be noted that, on model resolution, Zeppelin Station and Ny-Ålesund are in the same grid box. Differences in the model results between the two stations, shown in Figure 6, are only due to the different temporal availability of the measurements. Interestingly the model agrees slightly better with the observations at Zeppelin station, which is more exposed to long-range transport while Ny-Ålesund is often subject to a blocking situation that prevents mixing of air masses because of their respective location.





Figure 7 shows the second set of stations. Tiksi, Pallas and Utqiagvik (Barrow) show the same annual cycle as the stations in Figure 6, with high concentrations in winter and spring as well as minimum concentrations in summer. The model slightly underestimates BC at Tiksi in all months with high concentrations of over $50\,\mathrm{ng\,m^{-3}}$. For Pallas and Utqiagvik (Barrow) an overestimation by the model is found for January. Summit shows a different annual cycle in the observations, with the highest

median BC mass concentrations of slightly more than $30\,\mathrm{ng\,m^{-3}}$ observed in April and with slightly lower values in summer, and a minimum was observed for January. The model was neither able to reproduce this different annual cycle, nor the peak in the quartiles during September and December that were observed. However, the amount of BC mass agrees well between model and measurements, with values generally below $30\,\mathrm{ng\,m^{-3}}$.

Results for four Alaskan stations of the IMPROVE network are shown in Figure 8. There, the highest median BC concentra-

tions are observed in the summer months, with $70\,\mathrm{ng\,m^{-3}}$ at Gates of the Arctic in June, $50\,\mathrm{ng\,m^{-3}}$ at Trapper Creek in July, and with $40\,\mathrm{ng\,m^{-3}}$ and $60\,\mathrm{ng\,m^{-3}}$ in Tuxedni and Simeonof in August, respectively. The model noticeably fails to reproduce these summer maxima and instead produces the highest concentrations in January to March, in a similar way as for the other Arctic stations. In Tuxedni, the simulated median concentration lies at $60\,\mathrm{ng\,m^{-3}}$ while the observed one is at $10\,\mathrm{ng\,m^{-3}}$. The Brooks Range spans through Alaska from the Bering Sea in the West to the Beaufort Sea in the east with multiple peaks of more

than $2000\,\mathrm{m}$ above sea level. Situated south of Brooks Range, the four stations are shielded from Arctic. An underestimation of the orographic height in the coarse-resolved model could therefore be the reason for this misrepresentation.

The time correlation coefficient between measured and modeled BC mass concentrations for all available aerosol stations in the Arctic region is shown in Figure 9. Since pollution events in the Arctic can raise the BC concentrations to levels well above the background, the correlation coefficient is very sensitive to the model being able to reproduce the timing of pollution events.

Therefore, this analysis complements the median and quartiles discussed above.

The top right segment of each circle shows the correlation coefficient between the BCRUS model run and the measurements. Following clockwise are the correlations for the runs ACCMIP, ACCMIP-GFAS and ECLIPSE, respectively. The circle for Summit is not filled, since there the correlation coefficients are negative albeit close to zero (-0.06 for BCRUS). The negative correlation corresponds to the opposite annual cycle of surface BC in the BCRUS model results compared to the observations

as shown in Figure 7. For all other stations, correlation coefficients are positive. Simeonof on the Alaska Peninsula shows a very weak correlation with 0.09 for the different model runs. Tuxedni on the southern coast of Alaska also has a relatively low correlation coefficient of 0.44.

For the other Alaskan stations of the IMPROVE network, however, a correlation between observations and BCRUS model results is found that is robustly positive. Even for the stations where the annual cycle was not reproduced, the correct timing of

short term events leads to these positive correlation coefficients. Trapper Creek shows a correlation coefficient of 0.55, Denali National Park (NP) of 0.72 and Gates of the Arctic NP of 0.94. ACCMIP clearly performs worst of all experiments, with 0.14, 0.31 and 0.20, respectively, while the other runs do not differ strongly from each other. Taking the position and strength of actual biomass burning events into account is crucial for correctly reproducing the near-surface BC concentrations in Alaska.





The correlation coefficient at Oulanka is below 0.3 for all runs. This however is computed only on the basis of three months of measurements. The other European stations of Pallas, Ny-Ålesund and Zeppelin Station show also relatively low correlation coefficients of 0.45, 0.50 and 0.30 for BCRUS, respectively. The other runs behave similarly.

At the four northernmost stations, Tiksi, Utqiagvik (Barrow), Alert and Villum Research Station correlation coefficients of 0.55, 0.65, 0.60 and 0.60 are found for BCRUS, respectively. These four stations are located north of a big land mass, and likely show a good correlation, since concentrations are drastically different, when the wind is either coming from the land or the Arctic ocean. With the exception of Tiksi, the ACCMIP run does not produce considerably weaker correlations with the observations than the other runs. At Tiksi, the highest correlation coefficient would be expected for BCRUS, since BCRUS comprises the most recent and detailed emissions specifically for Russia. With 0.56 compared to 0.71 (ACCMIP-GFAS) and 0.61 (ECLIPSE) the correlation is however the lowest.

## 4.2 Vertical distribution of BC

The BC mass mixing ratio from airborne measurements are a valuable source of information about the vertical distribution of BC. However, because of the logistical difficulties and high costs the spatial and temporal coverage is quite sparse. The aircraft campaigns used in this study for model evaluation are described in detail in Section 2.5.2, their geographical operation area is presented in Figure 4, and they are listed in Table 3. Each measurement point is compared to the nearest grid cell from the model, resulting in one average profile per campaign and run. We group the campaigns based on season, resulting in at least one profile per season, with better coverage during spring and summer, with three and four campaigns, respectively.

### 4.2.1 Winter

For the winter months (December-January-February; DJF) only data from the HIPPO campaign is available, starting with the first deployment during January 2009. We consider only data points north of $60° $N. The area covered by HIPPO is indicated by the blue box in Figure 4. As shown in Figure 10, observed BC mass mixing ratios were highest near the ground. Everything below $950\,\mathrm{hPa}$ is removed from the plot, because of unrealistically high measured BC mass mixing ratios near the ground of over $450\,\mathrm{ng\,kg^{-1}}$ on average, that could not be reproduced by the model. Starting at $950\,\mathrm{hPa}$ the simulated profile of BCRUS is very similar to the observed vertical distribution. Model results and measurements show a decrease of BC with height, with the BCRUS run overestimating the BC mass mixing ratio above $900\,\mathrm{hPa}$ by a factor of about two. The ECLIPSE run produces almost the same profile, however the runs ACCMIP and ACCMIP-GFAS produce lower values that, while still higher, are closer to the observed profile. Since the emission of BC for these runs is only lower in Central Asia (see Table 2) this likely points toward an overestimation of the modeled transport to the Arctic, possibly caused by an underestimation of wet removal.

### 4.2.2 Spring

The observed and modeled profiles of BC mass mixing ratios from the ARCTAS spring campaign over the American Arctic (orange box in Figure 4) in April 2008 can be found in Figure 11a. Observations show high values near the ground with a BC



mass mixing ratio of over $40\,\mathrm{ng\,kg^{-1}}$ and a steep increase from there towards a pollution layer with a maximum of almost $200\,\mathrm{ng\,kg^{-1}}$ at $600\,\mathrm{hPa}$ height. BCRUS (in green) correctly places this layer but underestimates its strength. A second BC layer is centered at about $400\,\mathrm{hPa}$ height, with the mixing ratio gradually decreasing above. All model runs including actual fire emissions are well able to capture the placement of the aerosol layer, while the magnitude is underestimated by a factor

of up to three. This could be caused by too low emissions in the source region with a correctly predicted transport, or just an effect of the coarse resolution of the model resulting the emissions for the fire event being mixed over the grid boxes instead of being concentrated in a confined plume. The other runs using GFAS produce similar results, with ECLIPSE and BCRUS performing best. The ACCMIP run without daily fire emissions deviates most from the observations. This shows that this BC distribution was in fact largely caused by a biomass burning fire plume.

The averaged profile of the measured BC mass mixing ratio for the HIPPO-3 campaign over the Pacific in March/April 2010 is plotted in Figure 11b. It shows observed mixing ratios of $20\,\mathrm{ng\,kg^{-1}}$ near the surface. There is a local minimum at $880\,\mathrm{hPau}$ height. The highest mass mixing ratio of BC is found at heights around $520\,\mathrm{hPa}$. ECHAM-HAM is able to reproduce this profile well up to a height of about $650\,\mathrm{hPa}$ in all runs. From there the model underestimates the amount of BC up to the height of about $400\,\mathrm{hPa}$. Above, the model overestimates the amount of BC. The overestimation at uppermost levels is twice as high

in the ECLIPSE and BCRUS model runs. They likely overestimate the long-range transport from Southeast Asian or Russian pollution sources. Close to the ground, BCRUS and ECLIPSE are better able to reproduce the observed mass mixing ratio.

The ACLOUD campaign took place around Svalbard in May and June 2017 and therefore represents late spring and early summer. As can be seen in Figure 11c, the mixing ratio during the ACLOUD campaign were low, with observed mass mixing ratios of $4\,\mathrm{ng\,kg^{-1}}$ to $5\,\mathrm{ng\,kg^{-1}}$ near the ground. A maximum with $14\,\mathrm{ng\,kg^{-1}}$ was observed at $800\,\mathrm{hPa}$, above which the mass

mixing ratio declined with increasing altitude. ECHAM-HAM reproduced this averaged profile relatively well, only placing the maximum too high at a height of $650\,\mathrm{hPa}$, where the observations again decreased to $4\,\mathrm{ng\,kg^{-1}}$. This overshooting by ECHAM-HAM, at upper levels, is mainly found for the last flight on 16 June 2017 (not shown separately). This already hints to the tendency of ECHAM-HAM to overestimate upper-layer transport of aerosol in summer as described in the text below. Note that for ACLOUD only BCRUS results can be presented because of the timeliness of the measurements.

**4.2.3 Summer**

Results for the comparison between the ARCTAS summer campaign over the American Arctic in June and July 2008 and the model results from ECHAM-HAM are shown in Figure 12a. The averaged profile over the campaign shows an increase in the BC mass mixing ratio with increasing height up to a maximum of $26\,\mathrm{ng\,kg^{-1}}$ at the $300\,\mathrm{hPa}$ level. As discussed by Matsui et al. (2011), air masses during this campaign were influenced by biomass burning in East Russia. Most of the BC from these

fires however was quickly removed from the atmosphere by wet depositions by heavy rain close to the source region (Matsui et al., 2011). BCRUS produced a similar profile, with BC mass mixing ratio values very close to the observations up to $700\,\mathrm{hPa}$ height. Above this level, the model overestimates the amount of BC. This points toward a misrepresentation in the wet removal process, or possibly a too efficient vertical mixing/uplift of fire aerosol in the model. ACCMIP strongly differs from the other runs and observations producing much higher mixing ratios below $550\,\mathrm{hPa}$ height. Above $570\,\mathrm{hPa}$ the BC mass mixing ratios



modeled by ACCMIP, however, are much closer to the observations. At this height, Matsui et al. (2011) found elevated values in measured CO pointing toward an influence by biomass burning fires. ACCMIP agrees best with the measurements, because the observed fires that lead to the overestimation in the other runs are not present in the run. In this way it produces values that are similar to the observations where biomass burning aerosol was removed.

Observations from HIPPO-4 (June/July 2011) and model results are compared in Figure 12b. Modeled and observed mixing ratios are relatively low with the highest observed BC mass mixing ratio at just above $19\,\mathrm{ng\,kg^{-1}}$. In BCRUS this maximum is found at $820\,\mathrm{hPa}$ - much lower than observed ($620\,\mathrm{hPa}$). The modeled vertical extent of this pollution layer is also thinner than observed. The major difference are far too high BC amounts between 500 and $200\,\mathrm{hPa}$ in the model results for all emission setups. Noteworthy is also the difference between the runs of ACCMIP and ACCMIP-GFAS, with ACCMIP performing better

than the others runs in reproducing the pollution layer in the lower troposphere. The emissions from the actual fires in the GFAS emissions seem to not have reached the observed height, but instead mostly remained below $800\,\mathrm{hPa}$. The ACCMIP biomass burning emission coincidentally allowed ECHAM-HAM to reproduce a biomass burning influenced layer in the same height as observed. The fact that all runs that use GFAS produce the same profile, while the only run without it produces a different profile, shows that the BC profile at least up to a height of $300\,\mathrm{hPa}$ is mainly caused by fire emissions.

The profile plot for HIPPO-5 (August/September 2011) shows low observed and modeled mass mixing ratios throughout the atmosphere (see Figure 12c). The observations show the highest mass mixing ratio close to the surface with $9\,\mathrm{ng\,kg^{-1}}$ and a decrease towards $870\,\mathrm{hPa}$ to values just over $1\,\mathrm{ng\,kg^{-1}}$. The observed BC mass mixing ratio stays low at layers above. BCRUS produces lower BC mass mixing ratios near the surface and overestimates the amount of BC above $850\,\mathrm{hPa}$. ACCMIP is the only run producing significantly different BC mass mixing ratios from the other runs with strong overestimation throughout the

profile and BC mixing ratios of up to $34\,\mathrm{ng\,kg^{-1}}$ at a height of $930\,\mathrm{hPa}$. This strong overestimation is related to inappropriate biomass burning emissions in ACCMIP in this area.

Figure 12d shows the BC mass mixing ratio of the ACCESS campaign in June 2012 averaged over all flights with exception of the transfer flights. The observations show a decrease from the near-surface mixing ratios of $13\,\mathrm{ng\,kg^{-1}}$ to a layer of cleaner air $870\,\mathrm{hPa}$ ($5\,\mathrm{ng\,kg^{-1}}$). The modeled BC profiles show very high mixing ratios near the ground and the minimum mixing

ratios at $900\,\mathrm{hPa}$ and increasing mass mixing ratios with increasing altitude. The model shows a considerable overestimation between 800 and $400\,\mathrm{hPa}$. For the ACCESS campaign the model again shows the tendency to strongly overestimate the amount of BC at high altitudes.

### 4.2.4   Fall

The second mission of the HIPPO campaign measured BC layering over the Pacific during November 2009. The fall profile is

shown in Figure 13. The highest BC mass mixing ratio of up to $40\,\mathrm{ng\,kg^{-1}}$ was found near the surface with a quick decrease to $5\,\mathrm{ng\,kg^{-1}}$ just below $900\,\mathrm{hPa}$. Above that height there is a lofted BC layer around $420\,\mathrm{hPa}$ with $26\,\mathrm{ng\,kg^{-1}}$. The BCRUS run underestimates the mixing ratios at the surface by $14\,\mathrm{ng\,kg-1}$. The lofted BC layer is placed slightly too low between 850 and $470\,\mathrm{hPa}$. The amount of BC however is well matched. With increasing altitude the amount of BC increases quicker than in the observations. The other runs show a very similar vertical layering of modeled BC mixing ratio. ACCMIP and ACCMIP-GFAS





underestimate the pollution layer below $500\,\mathrm{hPa}$. Again, the BC mixing ratios are strongly overestimated above $280\,\mathrm{hPa}$, in particular in the runs ECLIPSE and BCRUS. This is probably due an overestimation in the upper-level, long-range transport of North American or Russian air pollution.

## 5  Direct aerosol radiative effects of BC

Any difference in the prescribed anthropogenic and biomass burning emissions affects the atmospheric burden, the vertical layering, and deposition of BC aerosol as shown before. The corresponding uncertainties of the direct aerosol radiative effect (DRE) of BC in the atmosphere, and that of BC in snow are explored using the calculation method described in Section 2.4. We consider the top of atmosphere (TOA) DRE to estimate the impact on the atmospheric radiative balance and therefore the Arctic climate. The effect at the surface (bottom of atmosphere, BOA) is considered mainly, because of the implications
on surface temperatures and sea ice melting. The multi-year average TOA DRE of atmospheric BC for the BCRUS run is shown for all-sky conditions (cloudy and non-cloudy) and the years 2005-2009 in Figure 14a. Positive values of more than $0.2\,\mathrm{W\,m^{-2}}$ are calculated across the whole Arctic, indicating a net energy gain for the Arctic climate system. Values of more than $0.4\,\mathrm{W\,m^{-2}}$ are reached over the Arctic Ocean and the Russian Arctic.

Since most of the effect results from the solar spectral range the DRE is stronger in summer and close to zero in winter. At
the surface, the DRE of atmospheric BC is negative, as shown in Figure 14e, due to the absorption of incoming solar radiation by BC in upper atmospheric layers, which reduces the amount of energy reaching the surface. This negative effect is however smaller for the Central Arctic Ocean than anywhere else in the Arctic, with $-0.05\,\mathrm{W\,m^{-2}}$ to $-0.1\,\mathrm{W\,m^{-2}}$.

The BC-in-snow albedo effect for all-sky conditions, is shown in Figures 14b and f, as 2005-2009 multi-year annual mean, for TOA and surface, respectively. The difference between TOA and surface is small and mainly caused by clouds. The effect is
largest in coastal Greenland with around $1\,\mathrm{W\,m^{-2}}$, where snow is present throughout the year. Over the temporarily sea-ice and snow covered Arctic Ocean the albedo effect varies around $0.2\,\mathrm{W\,m^{-2}}$, which compensates the negative DRE of atmospheric BC at the BOA. The sum of the DRE of BC in the atmosphere and snow is shown in Figure 14c and Figure 14g for the TOA and surface, respectively. Over the temporarily sea ice covered Arctic Ocean the BOA DRE of all BC (in snow and atmosphere) is slightly positive (around $0.1\,\mathrm{W\,m^{-2}}$), while the TOA DRE is strongly positive with values up to $1.9\,\mathrm{W\,m^{-2}}$. Over the Arctic
Ocean the DRE of atmospheric BC is in the range of the DRE considering all aerosol species (not shown), but smaller over the continents. The all-aerosol DRE at the TOA would therefore be negative if no BC were present in the Arctic atmosphere.

The difference between the model runs is used to estimate the emission-related uncertainty of the Arctic energy budget. Therefore difference of the total radiative effect at TOA (all sky conditions) of ACCMIP-GFAS minus BCRUS as shown in Figure 14h is analyzed. In the ACCMIP-GFAS run, the TOA net all-sky positive radiative effect of BC is lower by $0.1\,\mathrm{W\,m^{-2}}$
to $0.2\,\mathrm{W\,m^{-2}}$ over most of the Arctic region. At the surface the difference is smaller with values of $0.05\,\mathrm{W\,m^{-2}}$ less in ACCMIP-GFAS over most of the Arctic, with the exception of parts of Russia, as shown in Figure 14h. The more recent and transient emission data with local refinement, therefore, results in a considerably stronger climate forcing due to anthropogenic and biomass burning BC. This shows that the TOA DRE of BC is more sensitive to an increase in BC burden due to the



different emission setups, than the BOA DRE, since the net energy gain caused by the reduction of the snow albedo is canceled out to some degree by the shadowing effect of atmospheric BC.

We therefore conclude that, according to our best estimate, BC causes a net energy gain for the Arctic on the annual mean at TOA as well as BOA. The uncertainty with respect to the emissions setup is roughly 25 % for TOA and BOA, but stronger
in absolute values at TOA.

## 6    Summary and Conclusions

In this study, the representation of Arctic black carbon (BC) aerosol particles in the global aerosol-climate model ECHAM6.3-HAM2.3 is evaluated with respect to different emission inventories. As a reference BC measurements at Arctic sites and from aircraft campaigns are used comprehensively. By comparing the effects of different state-of-the-art BC emission inventories, an
uncertainty range of current model estimates of the Arctic atmospheric BC burden and the local direct radiative effect (DRE) of BC is quantified. The uncertainties are explored with a focus on three influencing factors: (1) The influence of temporally variable biomass burning emissions, (2) The importance of recent air quality policies and economical developments, (3) the potential improvements by regional refinements in Russian BC sources. This is achieved by comparing four different emission setups.

The run BCRUS represents the best estimate of global emissions. It is using anthropogenic emissions from the ECLIPSE emission dataset, in Russia the BC emissions of ECLIPSE are replaced with the higher resolved and more recent data from Huang et al. (2015). For the biomass burning emissions GFAS is used, which derives the location and amount of emitted gas and aerosol particles from satellite. The ECLIPSE run uses ECLIPSE emissions and GFAS emissions for the biomass burning emissions. For the ACCMIP run we use the anthropogenic part of the ACCMIP emissions, which are widely used. We fixed
the emissions to year 2000, not taking into account the recent economic changes and variable biomass burning emissions. ACCMIP does not consider gas flaring emissions. In the run ACCMIP-GFAS, the fixed year 2000 biomass burning emissions are replaced by dynamic real-time fire data from GFAS.

The comparison between ACCMIP and ACCMIP-GFAS is used to estimate the impact of temporally variable biomass burning emissions. ACCMIP-GFAS and BCRUS are used to quantify the impact of recent developments in air quality policies
and economical developments. The difference between ECLIPSE and BCRUS shows the impact of a regional refinement.

The variable biomass burning emissions, are not particularly important for the annual mean of the Arctic BC burden, but are crucial for reproducing high pollution events. The different assumptions on anthropogenic emission based on economic development and air quality policies, result in an uncertainty in BC burden of more than $50\,\mu\mathrm{g\,m^{-2}}$ over the Arctic Ocean, which is 20 % of the annual mean BC load. The regional refinements in Russia mainly change the BC burden in this region and
will improve the ability of the model to reproduce local measurements.

The near surface BC concentrations could be reproduced reasonably accurately by ECHAM-HAM in most cases. The exception from this are stations that are challenging because of their surrounding orography and the horizontal model resolution, namely: Summit, Ny-Ålesund and Zeppelin Station where ECHAM-HAM falsely produced similar peak concentrations in late





winter and early spring as for all other stations. The sensitivity to the different emission setups is low in the summer. This is a result of low local emissions near the measurement sites in all runs and a reduced long-range transport from the mid-latitudes, as well as more precipitation in the summertime Arctic.

In the months with high modeled concentrations the model shows a high sensitivity to the changing emissions for the stations closest to the Arctic ocean. The observed monthly median BC peak concentrations in Tiksi were underestimated by the model, the run BCRUS that includes the most accurate gas flaring emissions produced the best results. For other stations, e.g. in Barrow in February, BCRUS showed a stronger overestimation, than the other runs.

A similar pattern can be observed for Zeppelin Station, Ny-Ålesund, Villum Research Station and Alert. Higher emissions lead to higher concentrations with no significant changes in the pattern of the annual cycle. Overall, however, it is difficult to decide which emission setup provides satisfactory agreement with the aerosol observations for all cases. This means that the annual cycle of Arctic stations reproduced by ECHAM-HAM is mainly controlled by the transport. Changing the amount and location by using a different emission setup only modulates the amount of the BC concentrations, but unexpectedly does no affect the seasonality significantly.

The correlation coefficients of near surface concentrations are generally good. This points toward a good agreement in the timing, especially of observed peak events. These peaks are most often caused by biomass burning. The run ACCMIP is the only one that shows significantly smaller correlation coefficients, since the biomass burning emissions for that run are fixed and not prescribed on a daily basis from satellite observations.

The evaluation using a combination of aircraft campaigns shows that in general the vertical distribution is well reproduced by ECHAM-HAM. The model results looks best during spring. In summer BC is systematically overestimated by the model at heights above 500 hPa. This overestimation has been described for the AeroCom models before by Schwarz et al. (2013).

In one observed summer case, described by Matsui et al. (2011), the model correctly reproduced the time and height of a biomass burning layer. In reality, however, the BC was largely removed from the airmass by wet removal near the source region, which the model could not reproduce and instead overestimated the amount of BC. This leads us to the conclusion that a misrepresentation in the wet removal process in the model leads to this overestimation.

The ECHAM-HAM simulations show that over the Arctic Ocean the net (solar plus terrestrial) TOA DRE of atmospheric BC is positive, with an annual average of over $0.4\,\mathrm{W\,m^{-2}}$. The BC-in-snow albedo effect causes an additional energy gain for the Arctic system of around $0.2\,\mathrm{W\,m^{-2}}$ over the central Arctic. Locally larger effects are calculated for coastal Greenland. The BOA DRE is stronger than the shadowing effect of BC causing a net energy gain. The emission related uncertainty of DRE both at TOA and BOA is roughly 25 %.

Overall, the current model version of ECHAM6-HAM2 performs considerably better, than in a previous model intercomparison study (Schwarz et al., 2017). In particular the seasonality, but also the layering of BC aerosol in the Arctic has improved and is now better than the AeroCom average. Reducing the overestimation of upper-level BC concentrations would be a big improvement since this still causes large uncertainties in climate models and recent direct radiative forcing estimates. Here, especially the representation of wet scavenging and convective mixing needs to be improved, since it is the biggest BC sink in the Arctic.



*Code and data availability.* The code for ECHAM-HAM is available to the scientific community according to the HAMMOZ Software Licence Agreement though the project website: https://redmine.hammoz.ethz.ch/projects/hammoz The model data for used for the plots will be available on through the World Data Center PANGAEA.

*Author contributions.* JS performed the ECHAM-HAM simulations, collected emission data and in-situ measurement data from the providers,
prepared the emissions, performed the analysis and wrote the manuscript. BH provided support for the ECHAM-HAM simulations, suggested in-situ measurement data providers, and provided advice during the analysis and on the project design. JQ, MZ, AE, JB and RC provided support in writing and designing the manuscript. RC gave advice on the emission data setup. WTKH provided the code and advice on the BC in-snow parameterization for ECHAM-HAM. JB, AH, YK, AM, PRS, BW, and MZ provided in-situ measurement data and associated discussion. IT provided advice throughout the project design, setup, analysis and writing progress.

*Competing interests.* The authors declare that they have no conflict of interests.

*Acknowledgements.* We gratefully acknowledge the funding by the Deutsche Forschungsgemeinschaft (DFG, German Research Foundation) – Project Number 268020496 – TRR 172, within the Transregional Collaborative Research Center "ArctiC Amplification: Climate Relevant Atmospheric and SurfaCe Processes, and Feedback Mechanisms (AC)[3]" R. Cherian was supported by the DFG project under grant agreement 637230. IMPROVE is a collaborative association of state, tribal, and federal agencies, and international partners. US Environmental
Protection Agency is the primary funding source, with contracting and research support from the National Park Service. The Air Quality Group at the University of California, Davis is the central analytical laboratory, with ion analysis provided by Research Triangle Institute, and carbon analysis provided by Desert Research Institute. We thank the principal investigators Brent Holben, Ihab Abboud, Antti Arola, Vitali Fioletov, Laurie Gregory, Rigel Kivi, Lynn Ma, Norm O'Neill, Mikhail Panchenko, Piotr Sobolewski, John R. Vande Castle, Rick Wagener, the co-investigators Piotr Glowacki, Grzegorz Karasiski, Sergey Sakerin, and their staff for establishing and maintaining the AERONET
sites used in this investigation. The HIPPO 1-5 data was provided by NCAR/EOL under the sponsorship of the National Science Foundation (https://data.eol.ncar.edu/). We would also like to thank the German Climate Computing Center (Deutsches Klimarechenzentrum, DKRZ) for the computing time and their service. We especially thank the developers of ECHAM-HAM. The ECHAM-HAMMOZ model is developed by a consortium composed of ETH Zürich, Max Planck Institut für Meteorologie, Forschungszentrum Jülich, the University of Oxford, the Finnish Meteorological Institute, and the Leibniz Institute for Tropospheric Research, and managed by the Center for Climate Systems
Modeling (C2SM) at ETH Zürich.

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





**Table 1.** Arctic BC budget averaged for the years 2005–2015 in $\text{kt mon}^{-1}$ for BCRUS.

|           | Sedimentation + Dry deposition | Wet deposition | Emission | Deposition-Emission |
|-----------|-------------------------------|----------------|----------|---------------------|
| January   | 2.0   | 14.0  | 12.5  | 3.5  |
| February  | 3.3   | 15.1  | 11.7  | 6.4  |
| March     | 1.7   | 16.3  | 11.2  | 6.9  |
| April     | 0.9   | 19.8  | 10.5  | 10.2 |
| May       | 0.7   | 19.0  | 11.3  | 8.3  |
| June      | 1.0   | 23.0  | 20.5  | 3.5  |
| July      | 2.2   | 38.3  | 41.4  | -0.9 |
| August    | 1.7   | 30.9  | 25.5  | 7.0  |
| September | 1.3   | 18.2  | 10.9  | 8.7  |
| October   | 1.5   | 16.7  | 9.9   | 8.2  |
| November  | 1.8   | 14.9  | 11.0  | 5.7  |
| December  | 4.1   | 14.3  | 12.0  | 6.4  |
| Year/Sum  | 21.9  | 240.5 | 188.2 | 74.1 |

**Table 2.** Area-weighted totals of BC emissions from anthropogenic sources and biomass burning fires for the main source regions (as shown in Figure 1) averaged for the years 2005–2015 in $\text{kt yr}^{-1}$.

| Model Run   | North America | Europe | Russia | Central Asia |
|-------------|---------------|--------|--------|--------------|
| BCRUS       | 400 | 408 | 687 | 2981 |
| ECLIPSE     | 399 | 401 | 645 | 2983 |
| ACCMIP      | 450 | 538 | 578 | 2005 |
| ACCMIP-GFAS | 515 | 533 | 542 | 1997 |

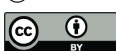



**Table 3.** Measurements overview. For aircraft campaigns the location of the airfield is given, unless no one base can be defined (denoted by *).

| | Latitude | Longitude | Period | Instrument/Inlet | Reference |
|---|---|---|---|---|---|
| Alert | 82.492° N | 62.508° W | 01-2012 - 12-2014 | Aethalometer/total | Backman et al. (2017b) |
| Pallas | 67.973° N | 24.116° E | | Aethalometer/total | |
| Tiksi | 71.973° N | 128.889° E | | Aethalometer/PM10 | |
| Summit | 72.580° N | 38.480° E | | Aethalometer/PM2.5 | |
| Zeppelin | 78.907° N | 11.889° E | | Aethalometer/total | |
| Ny-Ålesund | 78.927° N | 11.927° E | 04-2012 - 12-2015 | PSAP/PM10 | Sinha et al. (2017) |
| Barrow | 71.288° N | 156.792° W | 08-2012 - 12-2015 | PSAP/PM10 | |
| Villum | 81.600° N | 16.667° W | 05-2011 - 08-2013 | MAAP/total | Massling et al. (2015) |
| ACCESS campaign | 69.307° N | 16.118 ° E | 06-2012 | airborne SP2 | Roiger et al. (2015) |
| ARCTAS campaign | 64.821° N | 147.855° W | | airborne SP2 | Yutaka Kondo |
| HIPPO campaigns 1-5 | * | * | 01-2009 - 09-2011 | airborne SP2 | Wofsy et al. (2017) |
| PAMARCMiP campaign | 78.245° N | 15.492° E | 03-2017 | airborne SP2 | Herber et al. (2012) |
| ACLOUD campaign | 78.245° N | 15.492° E | 05-2017 - 06-2017 | airborne SP2 | Wendisch et al. (2018) |





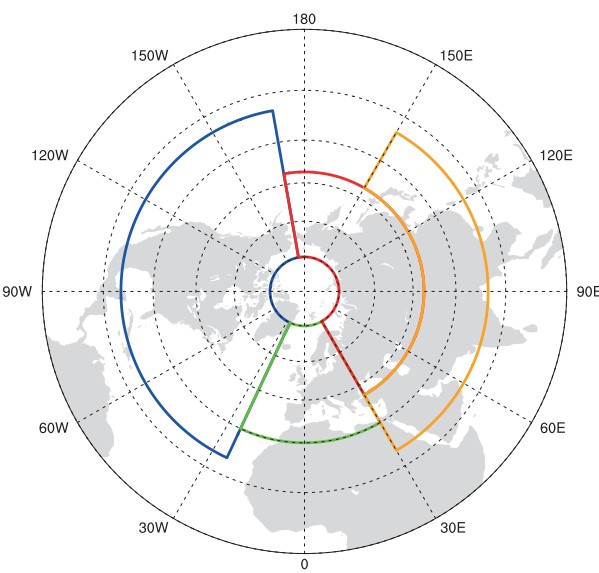

**Figure 1.** Regions indicate the area used for averaging presented in Table 2. North America in blue, Europe in green, Russia in red and Central Asia in orange.

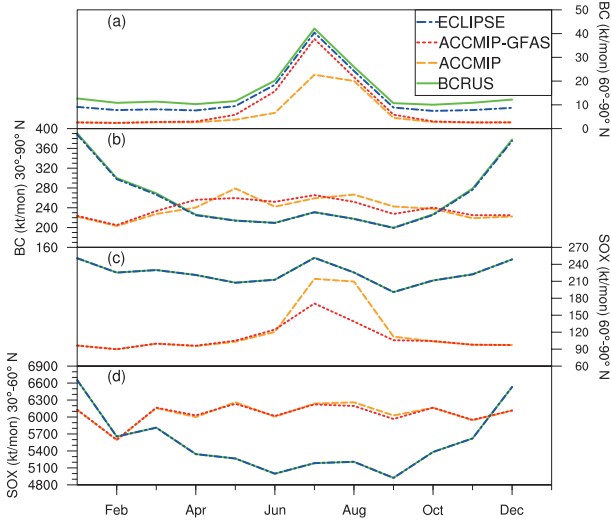

**Figure 2.** Multi-year monthly mean emissions of (a,b) BC and (c,d) SOx for the years 2005-2015. Values are integrated over the latitude band between 60° N and 90° N, and between 30° N and 60° N, respectively.



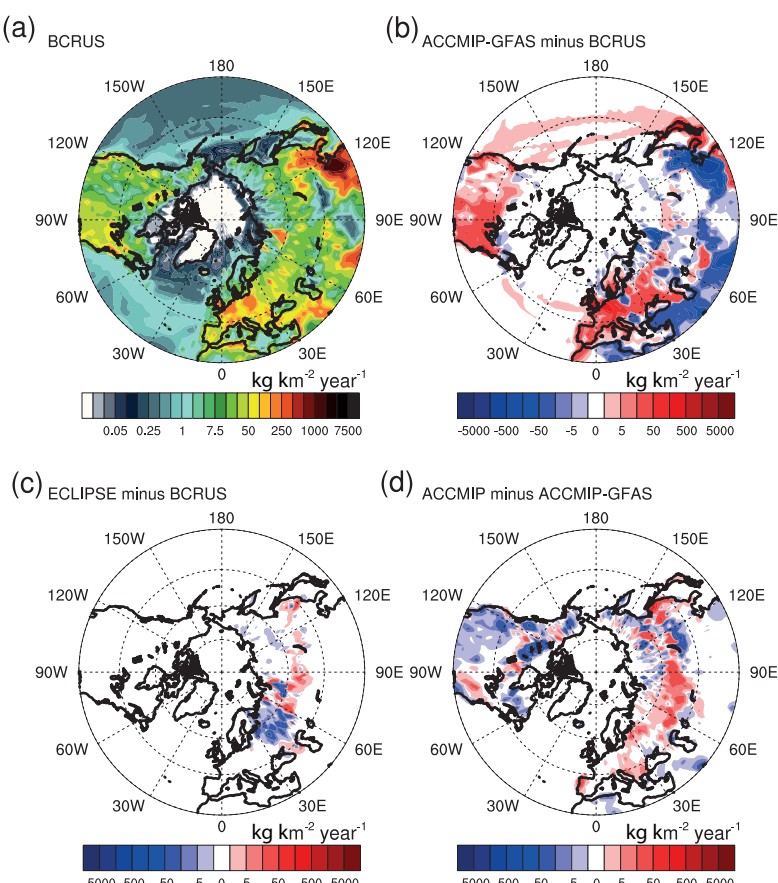

**Figure 3.** Maps of annual mean BC emissions for the years 2005-2015. (a) Absolute values are given for BCRUS. Difference between (b) the ACCMIP-GFAS and BCRUS results, (c) the ECLIPSE and BCRUS results and (d) between the ACCMIP and ACCMIP-GFAS results.





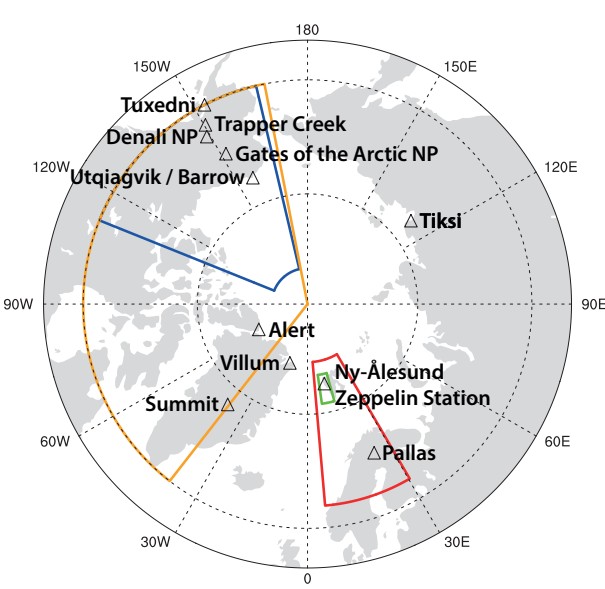

**Figure 4.** Geographic regions of Arctic aircraft campagins, the data of which is used for model evaluation: HIPPO in blue, ACLOUD and PAMARCMiP-2017 in green, ACCESS in red and ARCTAS in orange. Black triangles show the location of stations with BC surface measurements.



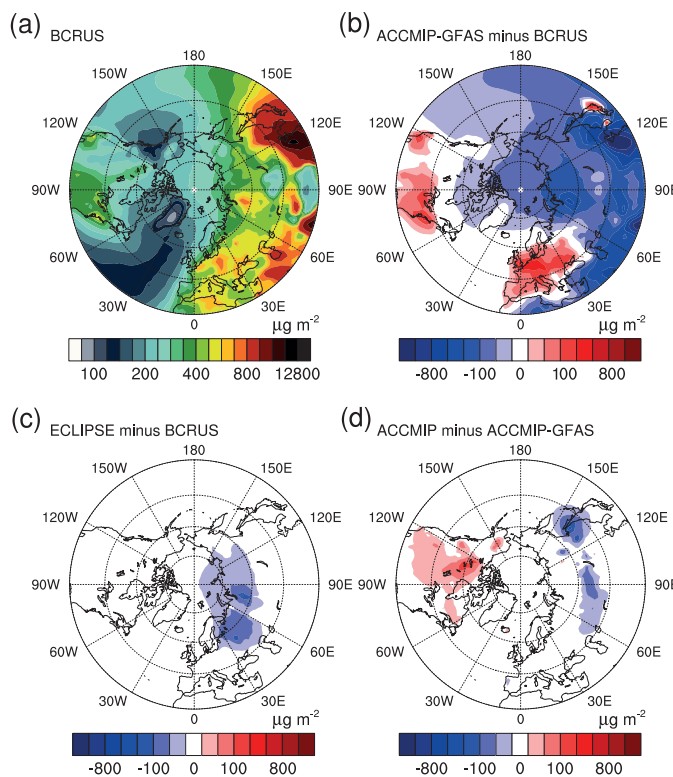

**Figure 5.** Contour plot showing the modelled atmospheric BC burden averaged over the simulation period (2005-2015). (a) Absolute values from the BCRUS setup, which is used as the reference. (b-c) Differences of ACCMIP-GFAS and ECLIPSE to the BCRUS run, respectively. Blue colors indicate lower BC burden than in the BCRUS run, red higher BC burden. (d) Difference in modelled atmospheric BC burden between ACCMIP and ACCMIP-GFAS.





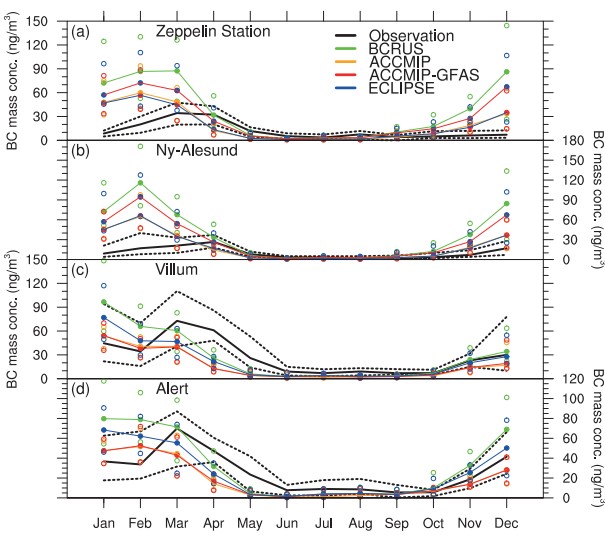

**Figure 6.** Near surface BC mass concentrations for Atlantic Arctic stations. Solid black line shows the multi-year monthly median BC mass concentration observed in (a) Zeppelin Station, (b) Ny-Ålesund, (c) Villum and (d) Alert. See Figure 9 for the geographical locations. Dashed black line indicates the observed upper and lower quartiles. In color the median different model runs with solid lines and filled circles, and the upper and lower quartiles with empty circles.

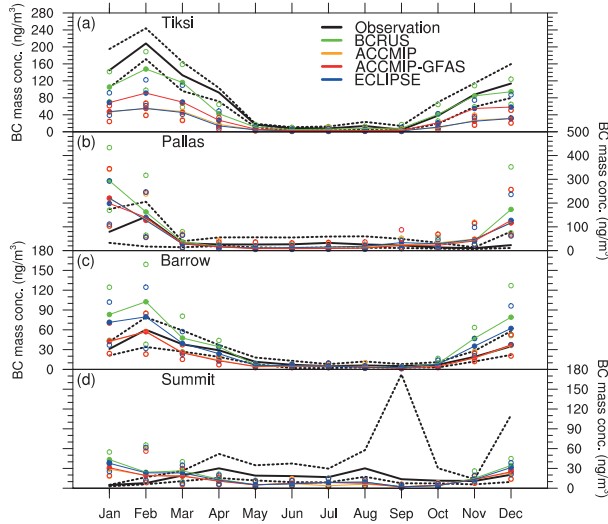

**Figure 7.** As Figure 6 for the stations in (a) Tiksi, Pallas, Barrow and Summit.





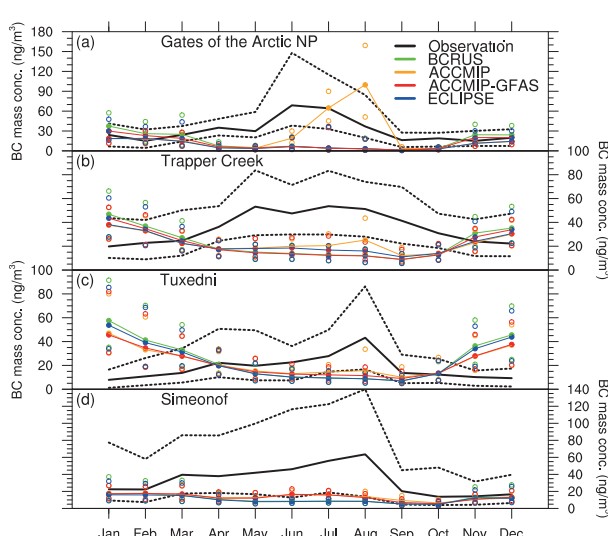

**Figure 8.** As Figure 6 for the American Arctic stations of the IMPROVE network in (a) Gates of the Arctic National Park, Trapper Creek, Tuxedni and Simeonof.




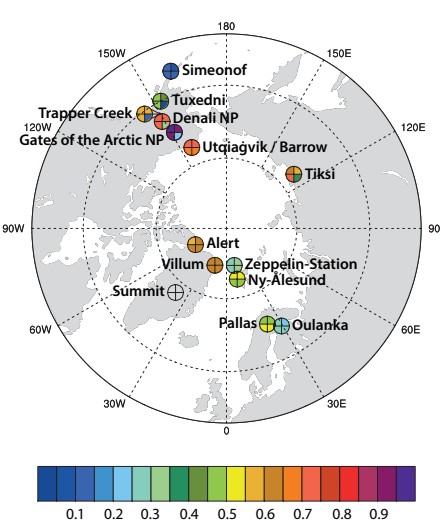

**Figure 9.** Map showing Arctic the sites where the near surface BC mass concentration was measured. Colors show the correlation coefficient between the measured and modeled daily averages. Correlation coefficients close to zero are not colored. Top right segment indicates the correlation coefficient for the BCRUS run, from there clockwise: ACCMIP, ACCMIP-GFAS, ECLIPSE runs, respectively. The label of Zeppelin Station is shifted to the north on the map for better visibility. The label of station Trapper Creek is shifted to the south east.

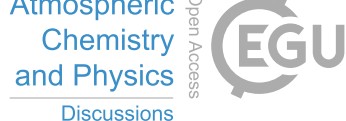



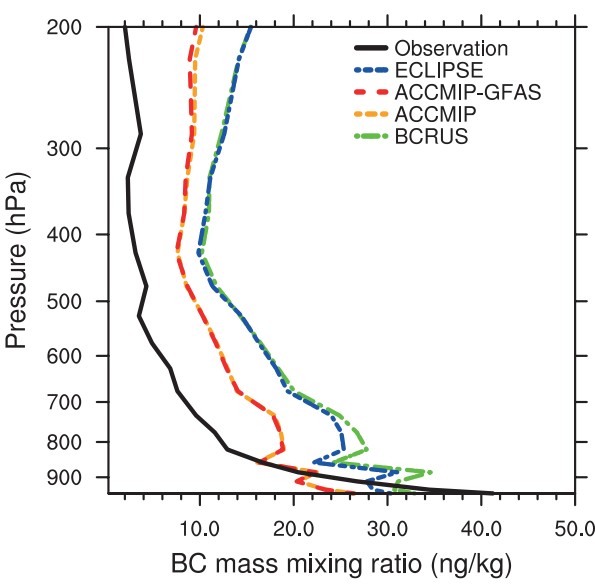

**Figure 10.** Vertical profiles of BC mass mixing ratios from airborne in-situ measurements during the flight campaign HIPPO-1 campaign in January 2009. The modeled BC mass mixing ratios were averaged over the vertical levels. The observations are shown in black, the different model runs are color coded.





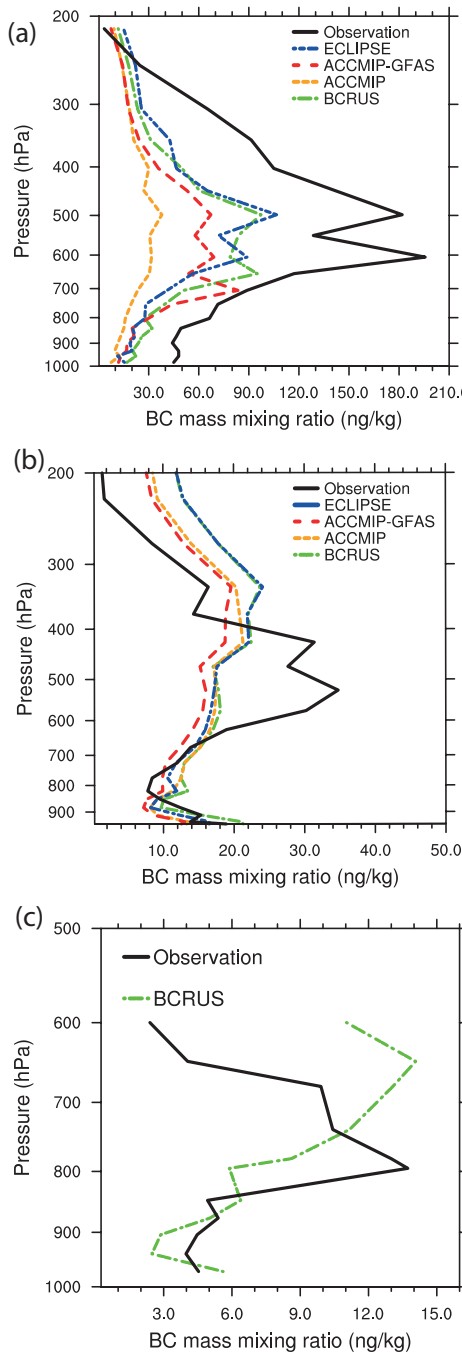

**Figure 11.** As in Figure 10, but spring campaigns (March/April/May). (a) ARCTAS spring campaign in April 2008. (b) HIPPO-3 campaign in March/April 2010. (c) ACLOUD campaign in May/June 2017. Note that model for 2017 are only available from the BCRUS run.





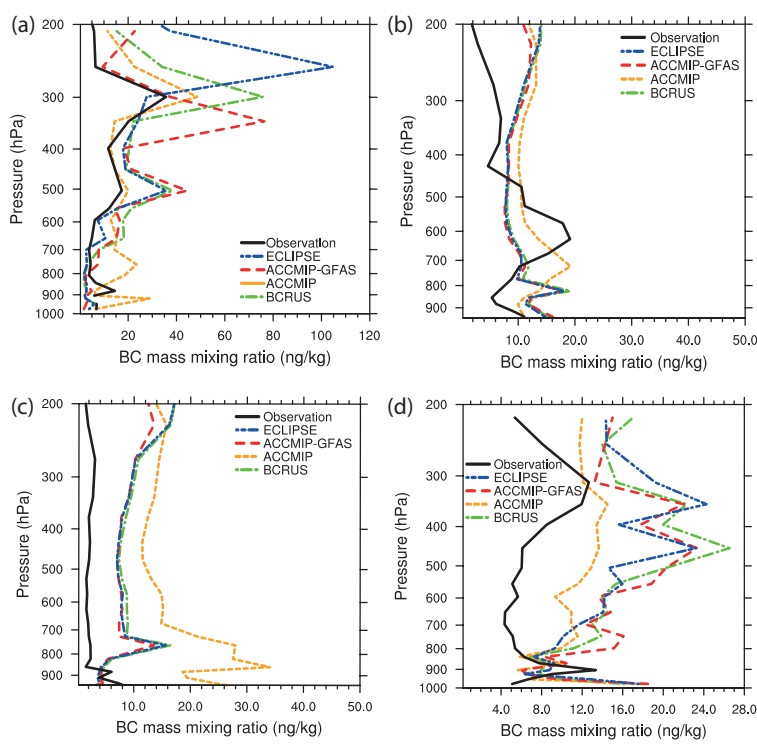

**Figure 12.** As in Figure 10, but summer campaigns (June/July/August). (a) ARCTAS summer campaign in June/July 2008. (b) HIPPO-4 campaign in June/July 2011. (c) HIPPO-5 campaign in August/September 2011. (d) ACCESS campaign in June 2012.





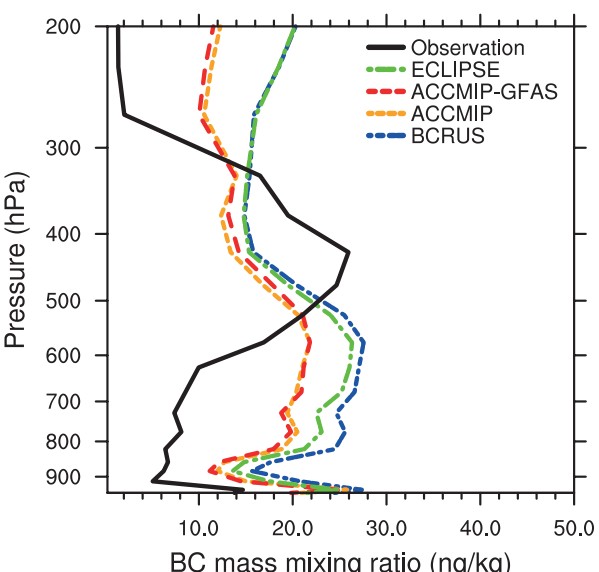

**Figure 13.** As in Figure 10, but the fall campaign HIPPO-2 in November 2009.



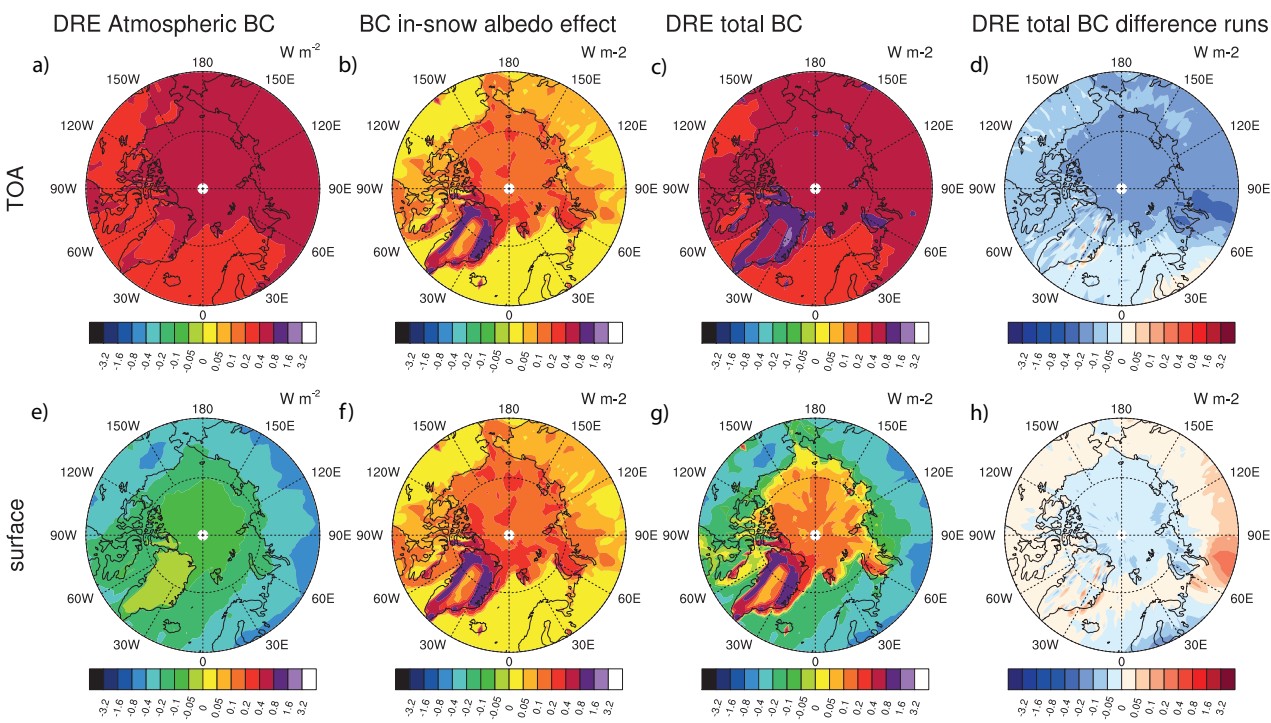

**Figure 14.** Multi-year mean all sky direct radiative effect (DRE) of BC for the period 2005-2009. Top row for TOA and bottom row for BOA. Panels (a) and (e) show the BCRUS net (terrestrial + solar) DRE of atmospheric BC. Panels (b) and (f) solar BC-in-snow albedo radiative effect. Panels (c) and (g) show the total of the radiative effects of atmospheric and BC deposited in snow (terrestrial plus solar). Panels (d) and (e) show the difference in the total BC radiative effect of the runs ACCMIP-GFAS minus BCRUS.