# Peer review of "The importance of the representation of air pollution emissions for the modeled distribution and radiative effects of black carbon in the Arctic"

_Atmospheric Chemistry and Physics, 2019_

## Referee Comment (RC1) · Anonymous Referee #1 · 3 Mar 2019

This study concerns modeling black carbon (BC) aerosols in the Arctic using the new version of the ECHAM-HAM aerosol model. Simulations of BC is compared with a comprehensive dataset of observations from surface stations and flight campaigns in the Arctic and mid latitudes. The authors run the model with four different emission datasets. All datasets are considered 'present-day', but there are substantial differences in the resulting BC burden and radiative forcing due to differences in the temporal representation of bio mass burning and the chosen emission year for anthropogenic emissions (2005 vs. 2017). This is an important finding. The paper is well-written and clear, and I recommend the manuscript for publication. I have a few minor comments below that I wish the authors could address.

[Figure]

In the introduction the authors present their study in a nice and clear way. Since the study concerns the representation of emissions; could you write a few more sentences on the other studies that have looked at the importance of emissions in representing BC in the Arctic?

L12, page 5, L18, page 6 and L15, page 17: I am a bit puzzled by the authors claiming that the Russian BC emissions from Huang et al 2015) is 'the best source available' and 'the best estimate of global emissions' just because it is a newer estimate compared to e.g. the ECLIPSE data set. The Huang et al chose an emission factor of 2.27 g/m3 in flaring compared to 1.6 g/m3 in ECLIPSE. In the paper, Huang et al admits their value is probably on the high end, and that there are no measurements in Russia confirming such high value being a representative average. It might be the case that the emissions are that high, but we do not know, and given the large uncertainties in estimating BC emissions from flaring, it is hard to say which data set is 'the best'. I am not suggesting that the authors should not use this data set as their base line, but I miss a discussion about uncertainties in the data sets and not just claim that one data set it the best.

L25-28, page 6: I am not sure if I understand this sentence? Contributing to layers?

L28, page 11: isn't also the difference because Zeppelin station is located on the mountain while the New Ålesund station is a ∼0 m altitude?

Figure 9: I like this figure, but it is possible to make the circles a bit bigger?

How did you compare the flight campaigns data with your model data in time?

L14, page 18: Can you add the numbers?

L30, page 18: (if possible) Is there any improvement(s) in the parameterization in the new model version you can highlight, that has led to better seasonality and vertical distribution of Arctic BC? I understand this is outside the scope of the study, but it would be interesting to know for other modelers if the information is available.

---

## Referee Comment (RC2) · Anonymous Referee #2 · 21 Mar 2019

The comment was uploaded in the form of a supplement:
https://www.atmos-chem-phys-discuss.net/acp-2019-71/acp-2019-71-RC2-supplement.pdf

---

## Referee Comment (RC3) · Anonymous Referee #3 · 29 Mar 2019

The authors present an evaluation of the ability for ECHAM-HAM to reproduce observed BC masses in the Arctic using different emissions datasets. There is large uncertainty in the representation of BC in the Arctic across general circulation models (GCMs) and part of this uncertainty is due to the availability of high-fidelity emissions data. This work aims to quantify that uncertainty and the consequent uncertainty in (surface and TOA) forcing. The authors use a large set of in-situ observations to evaluate the model performance and relate this to the shortcomings of each emissions dataset.

The only major concern I have with the work is in the scaling of the GFAS emissions

dataset. The original Kaiser et al. 2012 paper suggests a scaling factor of 3.4, but this has been found to be too high in some models and is somewhat arbitrary. The authors don't discuss this scaling factor at all, or what impact it could have on the results. Ideally another simulation with a lower (or no) scaling factor would be performed as a comparison, but the factor used and the resulting uncertainty should at least be discussed in both the technical setup and summary sections. Overall however this is a well written paper appropriate for publication in ACP after this, and the more minor points listed below, have been addressed.

- P1L2: 'positive effect' is a bit ambiguous, perhaps use 'positive (warming) effect'

- P2L12: 'conclusion, that' -> 'conclusion that'

- P3L3-4: 'are contributing' -> 'contribute'

- P4L7-8: This sentence is a quite difficult to understand. I believe only dust can be in the insoluble coarse and accumulation mode, but can also be in the soluble modes along with the other species. Perhaps include a table if that makes it clearer?

- P5L28-29: The claim that models still use ACCMIP 2000 emissions routinely should have a citation, e.g. Sand et al 2017.

- P7L9-10: This will include BC semi-direct effects then though. Perhaps make this clear and say something about the uncertainty this may introduce.

- Figure 2b: Axis label should read '60-90 N'

- P7L29-31: The MAC value of 9.8 m2 g-1 quoted from Zanatta et al. 2018 comes with a relatively large uncertainty (+/- 1.68), what uncertainty does this introduce into the BC concentration estimates? It should be discussed at least.

- P8L16: 'layering' -> 'vertical distribution'

- P8L16: This should probably be backed with a citation (e.g. Samset et al. 2013)

- P9L9: Typo in the 60 degrees Latex

- Fig 6: It's very hard the discern these small values, would a log scale be appropriate? The plots could perhaps also be made a bit bigger.

- P9L11: Avoid use of the colloquialism 'decent'. Perhaps 'sufficient'?

- P9L12: I would like to see a discussion here about the uncertainties introduced when comparing the in-situ aircraft measurements with a fairly coarse resolution global model (see e.g. Schutgens et al. 2016) and how this might affect your conclusions. Of particular relevance is Lund et al. 2018 who show that using monthly mean model output in comparison with aircraft BC measurements (and similar campaigns) can introduce significant biases.

- P11L6: '. . .atmosphere for reasonable. . .' -> '. . .atmosphere and hence reasonable. . .'

- P13L9-10: Could you hypothesise why this might be the case?

- Fig 9: This is a really nice diagram but it could definitely be bigger to make it clearer. The interpretation might also be easier if the correlation was indicated with a continuous color scale.

- P14L5-7: Be careful with the interpretation of these ARCTAS flights since they specifically went looking for plumes to sample and so you would expect a large sampling bias. This is implicit in some of the statements of Jacob et al. 2010.

- P16L3: This could also be due to an underestimation in removal as you've pointed out previously for other biases.

- P17L5: It's worth reiterating that this is an uncertainty due to emissions and doesn't account for the potentially large uncertainties due to removal.

- P18L5: ". . .by the model, the run. . ." -> ". . .by the model, but the run. . ."

- P18L24: I agree that wet removal seems a plausible cause for this bias, however

recent work with the same model (Lund et al. 2018) shows that it actually produces as good BC lifetime for the Arctic campaigns studied there so perhaps there are still transport issues.

- P18L31: "layering" -> "vertical distribution"

- P18L32: I can't see any comparison with the Aerocom average in the paper, this should be presented in the results or removed from the conclusion.

References

Schutgens, N. A. J., Gryspeerdt, E., Weigum, N., Tsyro, S., Goto, D., Schulz, M., and Stier, P.: Will a perfect model agree with perfect observations? The impact of spatial sampling, Atmos. Chem. Phys., 16, 6335-6353, https://doi.org/10.5194/acp-16-6335-2016, 2016.

Samset, B. H., Myhre, G., Schulz, M., Balkanski, Y., Bauer, S., Berntsen, T. K., Bian, H., Bellouin, N., Diehl, T., Easter, R. C., Ghan, S. J., Iversen, T., Kinne, S., Kirkevåg, A., Lamarque, J.-F., Lin, G., Liu, X., Penner, J. E., Seland, Ø., Skeie, R. B., Stier, P., Takemura, T., Tsigaridis, K., and Zhang, K.: Black carbon vertical profiles strongly affect its radiative forcing uncertainty, Atmos. Chem. Phys., 13, 2423-2434, https://doi.org/10.5194/acp-13-2423-2013, 2013

Marianne T. Lund, Bjørn H. Samset, Ragnhild B. Skeie, Duncan Watson-Parris, Joseph M. Katich, Joshua P. Schwarz, Bernadett Weinzierl : Short Black Carbon lifetime inferred from a global set of aircraft observations, npj Climate and Atmospheric Science 2018 https://doi.org/10.1038/s41612-018-0040-x

---

## Author Comment (AC1) · 31 May 2019

**Authors' Response to Reviewer's Comments**

Manuscript No.: acp-2019-71, submitted to APC
Title:          The importance of the representation of air pollution emissions for the
                modeled distribution and radiative effects of black carbon in the Arctic.
Authors:        J.Schacht, B.Heinold, J. Quaas, et al.

*We would like to thank the reviewers for their time and constructive comments, and hope that we have responded satisfactorily to all the points raised. The Authors' response and the changes in the revised manuscript are indicated by red and green text, respectively.*

**Anonymous Referee #1**

(1) Since the study concerns the representation of emissions; could you write a few more sentences on the other studies that have looked at the importance of emissions in representing BC in the Arctic?

(2) Agreed. In the Introduction, we added a brief review of previous studies on emission uncertainties, including Bond et al. (2004), Flanner et al. (2007), and Vignati et al. (2010).

(3) The following paragraph was added to the Introduction (L6, page 3): Bond et al. (2004) estimate the uncertainty in emission inventories to be a factor of about two. Flanner et al. (2007) conclude that for the climate forcing by BC in snow, the emissions introduce a bigger uncertainty than the scavenging by snow melt water and snow aging. However, the quality of emission inventories with models is difficult to assess, because of the dependence on the model (Vignati et al., 2010).

(1) L12, page 5, L18, page 6 and L15, page 17: I am a bit puzzled by the authors claiming that the Russian BC emissions from Huang et al 2015) is 'the best source available' and 'the best estimate of global emissions' just because it is a newer estimate compared to e.g. the ECLIPSE data set. The Huang et al chose an emission factor of 2.27 g/m3 in flaring compared to 1.6 g/m3 in ECLIPSE. In the paper, Huang et al admits their value is probably on the high end, and that there are no measurements in Russia confirming such high value being a representative average. It might be the case that the emissions are that high, but we do not know, and given the large uncertainties in estimating BC emissions from flaring, it is hard to say which data set is 'the best'. I am not suggesting that the authors should not use this data set as their base line, but I miss a discussion

about uncertainties in the data sets and not just claim that one data set it the best.

(2) Thank you for this comment. We changed Section 2.2. It now explains more clearly the dataset and, in particular, the information we gain from using ECLIPSE plus Russian BC emissions compared to ECLIPSE only. The idea of the sensitivity study is to span a reasonable uncertainty range for anthropogenic emissions. Accordingly, later in the text, the BCRUS run is referred to as an upper estimate.

(3) P5L9-14 changed to:
To address the importance of local emissions, we use the anthropogenic BC emission data set for Russian BC described in Huang et al. (2015). It is available for the year 2010 and originally is in 0.1 x 0.1 horizontal resolution, but has been scaled down to model resolution. Since the data set is limited to the area of Russia, we combine it with the ECLIPSE emission data. The Russian emissions are distributed between the different months with the monthly patterns of ECLIPSE.
The emissions of Russian gas flaring are more than 40% higher than in ECLIPSE resulting in part from a high conversion factor estimated for the fossil fuels found in Russia (Huang et al., 2015). It represents a reasonable yet high estimate of local emissions and is used as the reference setup. When compared, the ECLIPSE and Huang et al. (2015) emissions span an uncertainty range concerning gas flaring emissions.

P6L15 changed to:
This way, the BC sources are supposed to represent a high estimate, addressing the possibility of underestimation in the global datasets, in particular, with respect to gas flaring emissions.

P17L15 changed to:
The run BCRUS represents a recent estimate of global emissions with the special feature of a high estimate in local Arctic emissions especially of gas flaring.

(1) L25-28, page 6: I am not sure if I understand this sentence? Contributing to layers?

(2) Agreed. The sentence was rephrased for clarity.

(3) P6L25-28 changed to:
The transport efficiency from the East Asian sources to the Arctic is comparably low but the high emissions in this region makes it important for long-range transport to the Arctic upper troposphere.

(1) L28, page 11: isn't also the difference because Zeppelin station is located on the mountain while the New Ålesund station is at ~0 m altitude?

(2) We agree, this is the case in the real world but not in the model with its coarse horizontal resolution. Since both stations are less than 2 km apart, they are located in the exact same grid cell. Therefore, the modeled values are identical, except from the fact that the averaging periods differ because of the different availability of measurements.

(1) Figure 9: I like this figure, but it is possible to make the circles a bit bigger?

(2) Adjusted Figure 9.

(1) How did you compare the flight campaigns data with your model data in time?

(2) Thank you for making us aware that this was not properly described. The closest time step of the 12-hourly model output was selected for each data point in the flight campaign. This explanation was added to the methods' part.

(3) Inserted at P13L15:
Each measurement point is compared to the nearest grid cell *at the closest time step* from the model, resulting in one average profile per campaign and run.

(1) L14, page 18: Can you add the numbers?

(2) The analysis revealed correlation coefficients better than 0.45 for most stations while at Summit, Simeonof, Zeppelin and Oulanka the temporal correlation was poor with correlation coefficients below 0.3. The information is now given in the text.

(3) Inserted at P18L15:
The correlation coefficients of near surface concentrations are generally good, *with 0.45 and higher for most stations.* This points toward a good agreement in the timing, especially of observed peak events. These peaks are most often caused by biomass burning. *The exceptions are Summit, Simeonof, Zeppelin Station, and Oulanka, with correlation coefficients below 0.3.*

(1) L30, page 18: (if possible) Is there any improvement(s) in the parameterization in the new model version you can highlight, that has led to better seasonality and vertical distribution of Arctic BC? I understand this is outside the scope of the study, but it would be interesting to know for other modelers if the information is available.

(2) The now default wet removal scheme that is taking the wet aerosol size into account plays a big role. The scheme and its effect are discussed in Croft et al. (2010). We noticed this in a short test experiment.

(3) Insert at P18L19:
This improvement over older model versions is at least partly achieved with the aerosol side dependent wet removal scheme by Croft et al. (2010).

**Anonymous Referee #2**

(1) 1.I found quite a few typos in the text, most of which could be fixed using a simple spell checker.

(2) Our apologies for this. The manuscript was proofread carefully again to avoid typos in the revised version.

(1) 2.I think in Section 2.2 you could explain the different emission scenarios and the differences between them a little bit better. First off, if I'm not mistaken, the wildfire emissions in ACCMIP are decadal mean values based on GFEDv2, but they nevertheless have a monthly resolution. A big difference between ACCMIP and ECLIPSE is that the latter provides monthly varying emissions for many sectors, while ACCMIP does not. Monthly changing emissions should have an effect on the time evolution of the BC concentrations in the Arctic, especially close to the surface. It is not clear from the text whether the emissions by Huang et al. also provide monthly varying

emissions and, if not, how this has been dealt with when combining them with ECLIPSE. Furthermore, it should be noted that emissions with a high spatial resolution only provide limited improvements in the simulations here, as they anyway have to be re-gridded to a T63 resolution.

(2) Thank you for this well-informed comment. It made me realize that it was not properly discussed how the BCRUS emission were implemented. The methods section was extended as suggested.

(3) At P5L2 we added: […], linked to the Representative Concentrations Pathways (RCPs) for all later years (2000-2100) (Lamarque et al., 2010). *The anthropogenic emissions remain constant throughout the year. The biomass burning emissions vary monthly over the course of one year, but are only scaled by a factor between the years and do not differ in their location.*

At P5L6-8 the paragraph was extended: […] Unlike the ACCMIP emission data set, *the anthropogenic emissions are seasonally varying for the different sectors, and they* also include emissions from gas flaring. However, gas flaring emissions from Northern Russia, […], have been discussed *to be difficult to measure and therefore uncertain and possibly* too low in current emission inventories (Stohl et al., 2013).

*To address the importance of local emissions, we use the anthropogenic BC emission data set for Russian BC described in Huang et al. (2015). It is available for the year 2010 only and originally comes in 0.1° x 0.1° horizontal resolution, but here is interpolated to model resolution of T63 (~1.8°). Since the data set is limited to the area of Russia, we combine it with the ECLIPSE emission data. The Russian emissions are distributed between the different months with the monthly patterns of ECLIPSE.*
*The emissions of Russian gas flaring are more than 40% higher than in ECLIPSE resulting partly from a high conversion factor estimated for the fossil fuels found in Russia (Huang et al., 2015). It represents a reasonable yet high estimate of local emissions and is used as the reference setup. When compared, the ECLIPSE and Huang et al. (2015) emissions span a good uncertainty range concerning the uncertainty of gas flaring emissions.*

At P5L17 the sentence now reads: This allows for a representation of real-time fires in ECHAM-HAM **with daily changing emissions** and enables it to reproduce the biomass burning plumes […].

(1) 3. The procedure to calculate the DRE of BC should be explained in more detail (Section 2.4). Did you re-run the simulations without BC emissions, or leave out BC in the radiation calculations? If it was the latter, how was this done? The radiation code in ECHAM uses the aerosol wet diameter and an average refractive index of the aerosol particles (or rather the modes) to read out the optical properties from a pre-computed lookup table. The refractive index used is computed as volume-weighted average over all species in the particle. It therefore feels like one cannot just leave out one species, shouldn't you at least adjust the size of the particle (mode) accordingly?

(2) The run was performed with BC being emitted and transported, but skipped in the calculation of the refractive index. We re-evaluated this approach and noticed that the total particle number concentration was not adjusted to not include BC particles in the radiative code. This was now corrected and the manuscript was updated accordingly. The differences, however, are minor since the aerosol number of BC is small compared to the total number.
It is true, that the wet radius was not adjusted, since no separate (wet) radius is calculated for the individual species, but only for the seven aerosol modes. Adjusting for it would require major changes to the model, including specifically adding four additional tracers for the BC radii separated by mode. We agree that for future research this could be interesting.
Nonetheless, we still think that our approach is well suited to assess the radiative forcing of BC. In contrast, the approach of leaving BC out would cause non-linear effects on other aerosol species and their climate impact, making the runs hard to compare.

(3) Added at P7L10:
To calculate the DRE by BC, the ACCMIP-GFAS and BCRUS runs were repeated **with leaving BC out** in the *computation* of radiative fluxes. For this, BC was skipped in the calculation of the complex radiative index and the radiatively active number of particles while the wet radius of respective aerosol modes was not further adjusted. The DRE of BC is then derived from the difference of these

*two runs to their original setup. Note that with this method, the estimate includes the semi-direct effect of BC, which is not statically significant in the Arctic, as reported by Tegen and Heinold (2018). The DRE of BC is studied for the sub-period 2005-2009.*

(1) 4. In section 4.1, are the surface station data and the model data that you show collocated in a similar fashion as the aircraft data, or do you indeed show multi-year monthly averages. If the latter, did you constrain the model data to the years of the observations, or did you use the results of the entire model period?

(2) The collocation of station and model data was done in the same way as for the aircraft observations. For each measurement data point, the closest one in time and location was selected from the model output prior to any averaging.

(3) At P11L9 we added accordingly: *Each measurement point is compared to the nearest grid cell at the closest time step from the model. The medians are calculated after this sampling.*

(1) 5. On page 12, in the first paragraph, you discuss how the BC surface concentrations in Summit are so different from all other stations. I have done a plot similar to Figure 6 some years ago to evaluate ECLIPSE and ACCMIP against the same stations (not published) and asked the data providers about the same issue. It was suggested to me that the summer peak in Summit may be due to (local?) wild fire emissions, that might not be captured by observations. If this is the case, the model cannot really be blamed. Another issue is that Summit is situated at an altitude of over 3km, which may be much higher than the average orographic height of the model grid box in ECHAM. You could try correcting for that by evaluating the modelled BC concentrations in a model level that corresponds to this altitude.

(2) Thank you for sharing this experience. An underestimated influence of wildfires is possible and could explain the annual cycle. We are however not convinced that the emission inventory would be to blame. The summer 2017 fire events on the west coast of Greenland for example are shown by the GFAS emissions. The amount of BC emitted was however so low that it was barely noticeable in modeled atmospheric concentrations.

While the position of the Summit station is exceptional with 3207 m above sea level, the elevation of this area is represented in the model (with a surface geopotential of 32000 $m^{-2}$ $s^{-2}$).

(1)6.I agree that for model evaluation, where simulated concentrations can be compared to observational data with high temporal and spatial resolution, it is important that the emission inventories used also have a high resolution (both in time and space). This is especially true when one wants to improve how physical processes like, e.g. transport and deposition of aerosols, are modelled. However, when studying effects of changing aerosol emissions on climate, a lower resolution may be sufficient. Can you say anything about whether the monthly average arctic BC concentrations change qualitatively when using daily or monthly biomass burning emissions?

(2)For this study, we only used the daily GFAS emissions and the monthly fire emissions included in ACCMIP. Comparing the two data sets will most likely result more in a difference between constant and changing emissions. We however do agree that this comparison would be interesting. We would expect that the comparison of the monthly median values of the station measurements would give similar results, while correlations should be a lot worse. We would also expect the comparison to the aircraft campaigns to be worse.

(1)7.In the first paragraph of page 14 you briefly comment on the possibility that fire emissions may be artificially diluted in the relatively large model grid box, especially if the fire is small. Additionally to this, the way that fire emissions are inserted in the model may affect BC concentrations. If I am not mistaken, ECHAM distributes all wildfire emissions equally in vertical direction within the boundary layer. I think for monthly average emissions this is a good approximation, but for daily emissions this may lead to too fast vertical mixing. Therefore, thin fire plumes may be impossible to model correctly.

(2)Thank you for pointing this out. This is an important point that we will add in the discussion.

(3)Added at P5L18:
In ECHAM-HAM the biomass burning emissions are injected into the model boundary layer regardless of the actual injection height provided by GFAS, which is usually reasonable for most small and

moderate boreal fires while the injection height can be underesti-mated for specific events with high fire radiative power (Sofiev et al., ACP, 2009).

Added at P14L7:
[…] instead of being concentrated in a confined plume. In particu-lar, small local fire plumes may be too quickly diluted when emitted the boundary layer.

(1) 8.In Section 5, I think it would be helpful if you could give an arctic average TOA DRE, maybe in the form atmosphere+surface=total. In the abstract you state that the DRE is as high as 0.8Wm-2 – is this the yearly average for 60°-90°? Also, which scenario does this value correspond to?

(2) The value corresponds to the yearly average for the years 2005-2009 from the BCRUS setup but the 0.8 W m$^{-2}$ are a local effect. We will add an additional table to clarify this. And change the ab-stract accordingly additionally adding an Arctic average for 60°-90°N.

(3) At P1L12-13 we added to the abstract: Compared to commonly used fixed anthropogenic emissions for the year 2000, an up-to-date in-ventory with transient air pollution emissions results in *locally* up to 30% higher annual BC burden and an over *0.1* W m$^{-2}$ higher annual mean all-sky net direct radiative effect of BC at top of the atmos-phere *over the Arctic region (60°-90° N), with locally more than* 0.2 W m$^{-2}$ over the Eastern Arctic Ocean. We estimate BC in the Arctic to lead to an annual net gain of *0.5 W m$^{-2}$ averaged over the Arctic region, but locally* up to 0.8 W m$^{-2}$ by the direct radiative effect of atmospheric BC plus the effect by the BC-in-snow albedo reduction.

At P16L13 we inserted: Averaged over the Arctic (60-90° N), we es-timate the net DRE of atmospheric BC to 0.3 W m$^{-2}$ (see Table 5).

At P16L22 we added: On average the BC-in-snow albedo effect is 0.1 W m$^{-2}$.

At P16L26 we added: (-0.2 W m$^{-2}$ on spacial and annual average).

P16L30-31 we changed to: In the ACCMIP-GFAS run the TOA net all-sky positive radiative effect of BC is lower by 0.1 W m$^{-2}$ on the

regional average (60-90° N, see Table 5), but more than 0.2 W m$^{-2}$ higherregionally over the Barents and Kara Sea.

(1) 9. I see the point of all the panels in Figure 14 having the same data range, but on the other hand this makes it hard to see any features, especially in panels a and c. Also, do the numbers at the colour bars correspond to the centres of the coloured boxes or to the borders between them. In particular, which colour corresponds to zero?

(2) Agreed. We revised the plot to make it clearer.

Minor comments:

(1) 1. page 2, line 31: Do models really tend to over-estimate BC concentrations at the surface?

(2) Thank you for catching this. We definitely overstated this here and it should be made clear that at the surface this is only the case for some models, while most underestimate at the surface. We added two references that show this.

(3) P2L31 changed to: Despite a good agreement between BC obtained from models and observations close to source regions (Bond et al., 2013), in the remote Arctic regions, models *tend to predict a too low BC concentration at the surface in winter and spring while only some models overestimate it (Eckhardt et al. 2015). However, in the upper troposphere, models tend to overestimate the BC concentrations.*

(1) 2. section 2.3: How long was the spin-up of the simulations?

(2) The spin-up was 3 months.

(3) P5L22/23 changed to: The model simulations cover the 11-year period 2005-2015*, with a spin-up period of three months.*

(1) 3. page 6, lines 29--31: Could you try to re-formulate that sentence?

(2) Done.

(3) P6L29-31 changed to: *BC emissions from ACCMIP-GFAS are higher than those of BCRUS in North America, Europe, western*

*Russia and Japan. They are, however, lower in northern Russia by more than 3500 kg km$^{-2}$ yr$^{-1}$ less and China by more than 2800 kg km$^{-2}$ yr$^{-1}$ less, respectively.*

(1) 4.page 12, line 17: By time correlation, do you mean the Pearson correlation coefficient of the collocated data?

(2) Correct. It is the Pearson correlation coefficient that is considered here.

(3) P12L17 changed to: The *Pearson correlation between the collocated data of* measured and modeled mass concentrations for all available aerosol stations in the Arctic region is shown in Figure 9.

(1) 5.page 15, line 26: The last sentence in this paragraph seems quite redundant to me.

(2) We agree. Sorry about this. Removed.

(1) 6.page 18, lines 21--24: This may also be a resolution problem, as both the cloud and the smoke plume may not "fill" the entire grid box.

(2) This is a good point, which we are happy to include.

(3) After P18L21-24 we added: *In addition, even if the modeled amount of precipitation was correct, wet removal could be underestimated due to the resolution problem that both, smoke plume and precipitating cloud, do not fill the corresponding grid cell. This is a general problem when investigating specific small-scale events in a coarsely resolved model. While in general, the BC lifetime of ECHAM-HAM was discussed to be good (Lund et al. 2018), in this example, however, the model is incapable of reproducing the observations.*

**Anonymous Referee #3**

(1) The only major concern I have with the work is in the scaling of the GFAS emissions dataset. The original Kaiser et al. 2012 paper suggests a scaling factor of 3.4, but this has been found to be too high in some models and is somewhat arbitrary. The authors don't discuss this scaling factor at all, or what impact it could have on the

results. Ideally another simulation with a lower (or no) scaling factor would be performed as a comparison, but the factor used and the resulting uncertainty should at least be discussed in both the technical setup and summary sections. Overall however this is a well written paper appropriate for publication in ACP after this, and the more minor points listed below, have been addressed.

(2) We originally ran the model using GFAS emissions with the scaling factor 3.4, which led to a strong overestimation in the Arctic. Since this factor seemed arbitrary, it was discarded for all subsequent runs, which instead used the emissions without any scaling.

(3) Added at the end of Section 2.2:
*In previous works with ECHAM-HAM, GFAS emissions were often used with an emission factor of 3.4 as proposed by Kaiser et al. (2012). Since this led to a strong overestimation in BC concentrations in comparison to ground-based and airborne observations at mid and high-latitudes, the corresponding setup was discarded.*

At P17L22 in Section 6 we added:
*An emission factor of 3.4 that is commonly used for GFAS emissions (Kaiser et al., 2012) was not used since it led to a strong overestimation in BC concentrations of mid and high-latitudes.*

(1) - P1L2: 'positive effect' is a bit ambiguous, perhaps use 'positive (warming) effect'

(2) Changed.

(1) - P2L12: 'conclusion, that' -> 'conclusion that'

(2) Changed.

(1) - P3L3-4: 'are contributing' -> 'contribute'

(2) Changed.

(1) - P4L7-8: This sentence is a quite difficult to understand. I believe only dust can be in the insoluble coarse and accumulation mode, but can also be in the soluble modes along with the other species. Perhaps include a table if that makes it clearer?

(2) You are correct. We changed the sentence and added a table

(3) The hydrophobic Aitken mode contains BC and OC. In the hydrophilic Aitken mode, they are internally mixed with SU. The hydrophobic accumulation and coarse mode only contain DU. The hydrophilic accumulation and coarse mode contain BC, OC, DU and SS, all internally mixed with SU. See table X.

|  | Nucleation | Aitken | Accumulation | Coarse |
|---|---|---|---|---|
| Hydrophobic |  | BC, OC | DU | DU |
| Hydrophilic | SU | BC, OC, SU | BC, OC, DU, SU, SS | BC, OC, DU, SU, SS |

(1) - P5L28-29: The claim that models still use ACCMIP 2000 emissions routinely should have a citation, e.g. Sand et al 2017.

(2) We agree and have included references.

(3) References added at P5L28/29: ACCMIP emission data is still widely used for model experiments, *in some cases using the RCPs (Lund et al. 2018), or fixed for the year 2000 (Sand et al. 2017)*.

(1) - P7L9-10: This will include BC semi-direct effects then though. Perhaps make this clear and say something about the uncertainty this may introduce.

(2) Correct, it does include the semi-direct effects of BC. We addressed this now.

(3) Added at P7L10:
To calculate the DRE by BC, the ACCMIP-GFAS and BCRUS runs were repeated **with leaving BC out** in the *computation* of radiative fluxes. For this, BC was skipped in the calculation of the complex radiative index and the radiatively active number of particles while the wet radius of respective aerosol modes was not further adjusted. The DRE of BC is then derived from the difference of these two runs to their original setup. Note that w*ith this method, the estimate includes the semi-direct effect of BC, which is small on large-scale average since positive and negative effects cancel each other out and is not statically significant in the Arctic (*Tegen and

Heinold, 2018). *The DRE of BC is studied for the sub-period 2005-2009.*

(1)- Figure 2b: Axis label should read '60-90 N'

(2) Fixed.

(1) P7L29-31: The MAC value of 9.8 m2 g-1 quoted from Zanatta et al. 2018 comes with a relatively large uncertainty (+/- 1.68), what uncertainty does this introduce into the BC concentration estimates? It should be discussed at least.

(2) The uncertainty in the MAC value of +/-1.68 relates to an uncertainty in the BC concentration of -21% to +15%.

(3) Added after P7L31: Zanatta et al. (2018) give an *uncertainty of the MAC value of +/- 1.68 $m^2$ $g^{-1}$. This implies an uncertainty range of approximately -20% to +15% for the observed BC concentrations.*

(1)- P8L16: 'layering' -> 'vertical distribution'

(2) Changed.

(1)- P8L16: This should probably be backed with a citation (e.g. Samset et al. 2013)

(2) The suggested reference was added to the revised manuscript.

(1) P9L9: Typo in the 60 degrees Latex

(2) Corrected.

(1) Fig 6: It's very hard the discern these small values, would a log scale be appropriate? The plots could perhaps also be made a bit bigger.

(2) We have seen this type of plot with log scales before, but have never been a fan of that. Concerning the size, we are bound by the rules of the journal and fail to come up with an idea of how to use the given space more effectively. We are sorry to not follow your advice here.

(1) - P9L11: Avoid use of the colloquialism 'decent'. Perhaps 'sufficient'?

(2) Thank you. We were not aware the word is inappropriate here.

(3) At P9L11 the sentence was changed accordingly: Even though aircraft campaigns can only give information within a short time window, the combination of different campaigns allows to cover *the almost entire year* except from December, February, September and October, the months for which no aircraft data is available.

(1) - P9L12: I would like to see a discussion here about the uncertainties introduced when comparing the in-situ aircraft measurements with a fairly coarse resolution global model (see e.g. Schutgens et al. 2016) and how this might affect your conclusions. Of particular relevance is Lund et al. 2018 who show that using monthly mean model output in comparison with aircraft BC measurements (and similar campaigns) can introduce significant biases.

(2) In the text, we describe that we do not use monthly mean model output for any comparison but always the time and space collocated 12-hourly instantaneous output, which is then averaged in the respective months. Obviously, this was not clear enough.
Anyway, we think that a general discussion about the problems of the comparison would be beneficial and fit well. We added it together with information on how the comparison with the model was done. Thank you making me aware of the Lund et al. 2018 paper.

(3) Added at P9L12: *The comparison between a coarsely resolved model and aircraft measurements is challenging because of many factors. Any observed features of subscale lifetime or spatial extend will be missed or at least underestimated by a model that is designed to estimate climate relevant effects over multiple years. Schutgens et al. (2016) suggest either spatio-temporal averaging of both measurements and spacial interpolated model data or increasing the model resolution to achieve best agreement. Lund et al. 2018 show that using only monthly mean model output introduces significant biases.*
*In this study, we sample from the models twelve-hourly output for each measurement point during one campaign before averaging to one vertical profile, without prior interpolation.*

(1) - P11L6: '...atmosphere for reasonable...' -> '...atmosphere and hence reasonable...'

(2) Changed.

(1) - P13L9-10: Could you hypothesise why this might be the case?

(2) We assume that the lowest correlation coefficient for BCRUS at Tiksi is caused by the strength of peak concentrations above background caused by a pollution event at the correct time. The monthly medians are lower for ACCMIP-GFAS than for ECLIPSE, which in turn is lower than BCRUS (see Fig. 6a). Therefore, a correctly predicted pollution event increases the correlation more for ACCMIP-GFAS.

(1) - Fig 9: This is a really nice diagram but it could definitely be bigger to make it clearer. The interpretation might also be easier if the correlation was indicated with a continuous color scale.

(2) The size of Figure 9 was increased. We agree interpretation might be easier, but we prefer to keep the strong contrast over the continuous scale for good readability.

(1) - P14L5-7: Be careful with the interpretation of these ARCTAS flights since they specifically went looking for plumes to sample and so you would expect a large sampling bias. This is implicit in some of the statements of Jacob et al. 2010.

(2) Thank you for making us aware. We did not pay enough attention to the mentioned caveat, which we are happy to include in the revised manuscript.

(3) P14L5-7 we added: "This could be caused by too low emissions […] or just an effect of the coarse resolution of the model resulting the emissions for the fire event being mixed over the grid boxes instead of being concentrated in a confined plume. *In addition, there is the possibility of a large sampling bias with fire plumes being specifically probed during the campaign (Jacob et al. 2010).* The other runs using GFAS produce similar results, […]"

(1) - P16L3: This could also be due to an underestimation in removal as you've pointed out previously for other biases.

(2) Agreed. This was added here again.

(3) P16L3 extended to: Again, the BC mixing ratios are strongly overestimated [...]. This is *either* due to an overestimation in the upper-level, long-range transport of North American or Russian air pollution, or *by an underestimation in removal which could contribute to the upper-level transport.*

(1) - P17L5: It's worth reiterating that this is an uncertainty due to emissions and doesn't account for the potentially large uncertainties due to removal.

(2) Thank you for this comment. As this is one of main messages of the study we will stress it here again.

(3) Added at P17L5: The uncertainty with respect to the emissions setup is roughly 25% for TOA and BOA, but stronger in absolute values at TOA. This is solely due to the uncertainties in emission, potential uncertainties in removal shown in the evaluation with observations are not included.

(1) - P18L5: "...by the model, the run..." -> "...by the model, but the run..."

(2) Corrected.

(1) - P18L24: I agree that wet removal seems a plausible cause for this bias, however recent work with the same model (Lund et al. 2018) shows that it actually produces as good BC lifetime for the Arctic campaigns studied there so perhaps there are still transport issues.

(2) As stated above, we were unaware of said paper and will cite it here. Thank you for making us aware. However, we am sure that while the BC lifetime over the whole model and over long enough timespans can be good, in this specific example the problem would always occur merely on basis of the resolution problem with both the precipitation, the aerosol plume and the overlap of both.

(3) P18L21-24 we rewrote: *In one summer case of an observed wet removal affected biomass burning plume, described by Matsui et al. (2011), the model correctly reproduced the time and height of the*

*biomass burning layer. It is known, that reproducing individual pollution events in exactly the correct way is impossible for a global model with this resolution, because both the aerosol transport, as well as the wet removal are affected by subscale processes. ECHAM-HAM overestimated the BC concentrations, because of this issue. While here the BC lifetime was overestimated, in general, the BC lifetime of ECHAM-HAM was discussed to be reasonably good (Lund et al., 2018).*

(1) - P18L31: "layering" -> "vertical distribution"

(2) Changed.

(1) - P18L32: I can't see any comparison with the Aerocom average in the paper, this should be presented in the results or removed from the conclusion.

(2) Agreed. Removed the phrase in question.